behaviour

birdsong, vocal mimicry, sequence analysis, lyrebird, social transmission, repertoire complexity

**Author for correspondence:**
Fiona Backhouse
e-mail: f.backhouse@westernsydney.edu.au

# Higher-order sequences of vocal mimicry performed by male Albert's lyrebirds are socially transmitted and enhance acoustic contrast

Fiona Backhouse[1], Anastasia H. Dalziell[1,2,3], Robert D. Magrath[4] and Justin A. Welbergen[1]

[1]The Hawkesbury Institute for the Environment, Western Sydney University, Richmond, NSW, Australia
[2]Centre for Sustainable Ecosystem Solutions, School of Earth, Atmospheric and Life Sciences, University of Wollongong, Wollongong, NSW, Australia
[3]Fuller Evolutionary Biology Program, Cornell Lab of Ornithology, Cornell University, Ithaca, NY, USA
[4]Research School of Biology, The Australian National University, Canberra, ACT, Australia

FB, 0000-0001-9308-620X; AHD, 0000-0003-3602-0495; RDM, 0000-0002-9109-609X; JAW, 0000-0002-8085-5759

Most studies of acoustic communication focus on short units of vocalization such as songs, yet these units are often hierarchically organized into higher-order sequences and, outside human language, little is known about the drivers of sequence structure. Here, we investigate the organization, transmission and function of vocal sequences sung by male Albert's lyrebirds (*Menura alberti*), a species renowned for vocal imitations of other species. We quantified the organization of mimetic units into sequences, and examined the extent to which these sequences are repeated within and between individuals and shared among populations. We found that individual males organized their mimetic units into stereotyped sequences. Sequence structures were shared within and to a lesser extent among populations, implying that sequences were socially transmitted. Across the entire species range, mimetic units were sung with immediate variety and a high acoustic contrast between consecutive units, suggesting that sequence structure is a means to enhance receiver perceptions of repertoire complexity. Our results provide evidence that higher-order sequences of vocalizations can be socially transmitted, and that the order of vocal units can be functionally significant. We conclude that, to fully understand vocal behaviours, we must study both the individual vocal units and their higher-order temporal organization.

## 1. Background

Human language and music are fundamentally composed of higher-order sequences: phonemes are organized into words, which form sentences that can be structured into narratives [1,2], and notes are organized into motives and phrases, which in turn are organized into songs and other complex musical compositions [3]. Likewise, many forms of animal communication occur in extended sequences of smaller behavioural units [4–6]. This is especially true for avian acoustic communication, where vocalizations are organized hierarchically as elements or syllables within songs, and songs within song bouts (or 'song sequences') [7,8]. While many studies treat songs as individual units of analysis, the higher-order organization of songs into sequences may too encode biologically relevant information, beyond the individual units [4]. However, the proximate and ultimate causes underlying the organization of sequences of songs remain unclear in all but a handful of species [4,9,10].

The temporal organization of sequences can provide insight into how the behaviour is acquired [11,12]. Sequence structures may be unique to individuals [13–15], possibly reflecting variation among individuals in vocal learning pathways [12,15]. Alternatively, sequence structures may be socially transmitted between individuals [11]. Social transmission of higher-order song sequences has been demonstrated in humpback whales [16,17], but has thus far only been suggested in songbirds, where the evidence is limited to shared transitions between song types among neighbouring individuals [11], shared themes or song packages [12,18] or the copying of tutor sequences under controlled laboratory conditions [19]. Investigating the distribution of sequence variants among wild individuals is important for discerning how birds perceive and learn complex, hierarchically organized behaviours beyond individual songs.

The temporal organization of sequences can have functional significance. Sequence organization can function in intrasexual communication, such as signalling intent to initiate combat [20], or in intersexual communication, such as signalling of quality. One common way birds signal individual quality to prospective mates is through repertoire complexity, the number and acoustic diversity of syllable or song types in an individual's repertoire [21–23]. Therefore, organizing song sequences in a way that enhances receiver perception of complexity would probably benefit reproductive success [24,25], particularly where receivers are under a time constraint for assessing male quality [14,26]. Repertoire complexity may be showcased by singing with little to no immediate repetition of song types (immediate variety), which maximizes the perceived repertoire size [27] and may increase the attention time of the receiver [15,28–30]. Individuals may further increase the efficacy of signalling repertoire complexity by organizing song sequences such that there is a high acoustic contrast between consecutive units [10,28,31], just as a high contrast increases the efficacy of visual signals [25]. While there has been much attention on the functional significance of repertoire size in birds [32], surprisingly little attention has been paid to whether and how birds organize their song sequences to enhance receiver perceptions of repertoire complexity.

One vocal behaviour in which sequences are especially overlooked is vocal mimicry. Mimetic song used in mate attraction is often highly varied and delivered in long bouts [33], creating substantial potential for higher-order organization of units into structures. Northern mockingbirds (*Mimus polyglottos*) are the only species in which the temporal organization of mimetic units has been investigated, and were found to cluster mimicry of the same or similar species more than expected by chance [34] with gradual acoustic changes between consecutive units [35]. Heterospecific imitations are reported in 11–15% of all songbirds [36], yet to our knowledge neither the mode of transmission nor the functional significance of the structure of mimicry sequences have been addressed.

Here, we investigate the organization, transmission and function of sequences of mimicry sung by male Albert's lyrebirds (*Menura alberti*). The Albert's lyrebird is a large oscine passerine known for its exceptional mimicry that appears to be largely used in the context of sexual display [37]. Males mimic vocally a number of multi-element avian sounds, both vocal and non-vocal (e.g. wingbeats), in a complex audio-visual display. Locality-specific, repeatable sequences

of mimetic units have been anecdotally reported in previous research [38]; however, this has never been tested formally, and the function of such structured sequences is unknown. Furthermore, there is some evidence that males learn individual mimicked sounds at least in part from other lyrebirds, with a study showing that while lyrebird males match the local dialect of the satin bowerbird (*Ptilonorhynchus violaceus*; a preferred model species), indicating some level of learning from heterospecifics, lyrebird copies are more acoustically similar to neighbouring lyrebirds than to the model sounds [39]. Thus, it seems plausible that male Albert's lyrebirds share sequences of vocal mimicry through social transmission as well.

To investigate sequences of mimicry of Albert's lyrebirds, we first determined the degree of repeatability in sequences sung by individuals. Next, to examine whether Albert's lyrebirds share sequences of vocal mimicry through social transmission, we determined the similarities in sequences within and between populations. Social transmission predicts high sequence similarities between individuals from the same population, with lower similarities between populations. Finally, to examine whether sequences are structured to enhance the perception of repertoire complexity, we tested for both immediate variety and acoustic contrast in the order of mimetic units within sequences. If sequences are organized to signal repertoire complexity, then the mimetic units will be sung with immediate variety and acoustic contrast between consecutive mimetic units will be high in all populations.

## 2. Methods

### (a) Study species and sites

Albert's lyrebirds are large (approx. 930 g), sedentary oscine passerines confined to the montane rainforest and wet sclerophyll forest in Bundjalung Country, eastern Australia [37,38]. Individual males are territorial and largely solitary during the breeding season, except during sexual interactions or territorial encounters [37]. Males differ from females and juveniles by their longer, more extravagant tail including highly filamented feathers [37]. During the breeding season between March and August, male Albert's lyrebirds perform dance-like displays in conjunction with their own song and sequences of vocal mimicry of other species in performances lasting from several minutes to over an hour [37,38]. Males display in 'dispersed leks' [38], with approximately 300 m between neighbouring individuals (F.B. 2019, unpublished data). Males have no role in parental care, and so display components may be learnt from both related and unrelated individuals. Females also sing and are capable of accurate vocal mimicry, although they do not mimic as extensively as males [37] (F.B. 2019, personal observation). We studied Albert's lyrebirds at five sites that encompass the species' range (see electronic supplementary material, figure S1 for map): Koonyum Range in Mt Jerusalem National Park (28.53° S, 153.40° E), Border Ranges National Park (28.38° S, 153.08° E), Binna Burra (within Lamington National Park; 28.21° S, 153.19° E), Tamborine National Park (27.93° S, 153.19° E) and the Goomburra section of Main Range National Park (27.97° S, 152.39° E).

### (b) Male song

Male Albert's lyrebirds have a varied but structured repertoire including species-specific 'whistle song' [40] and 'gronking' [38], as well as mimicry of heterospecific vocalizations and environmental sounds [37]. Here, we focus on the most common form of vocal mimicry ('sequential song' [38]), which

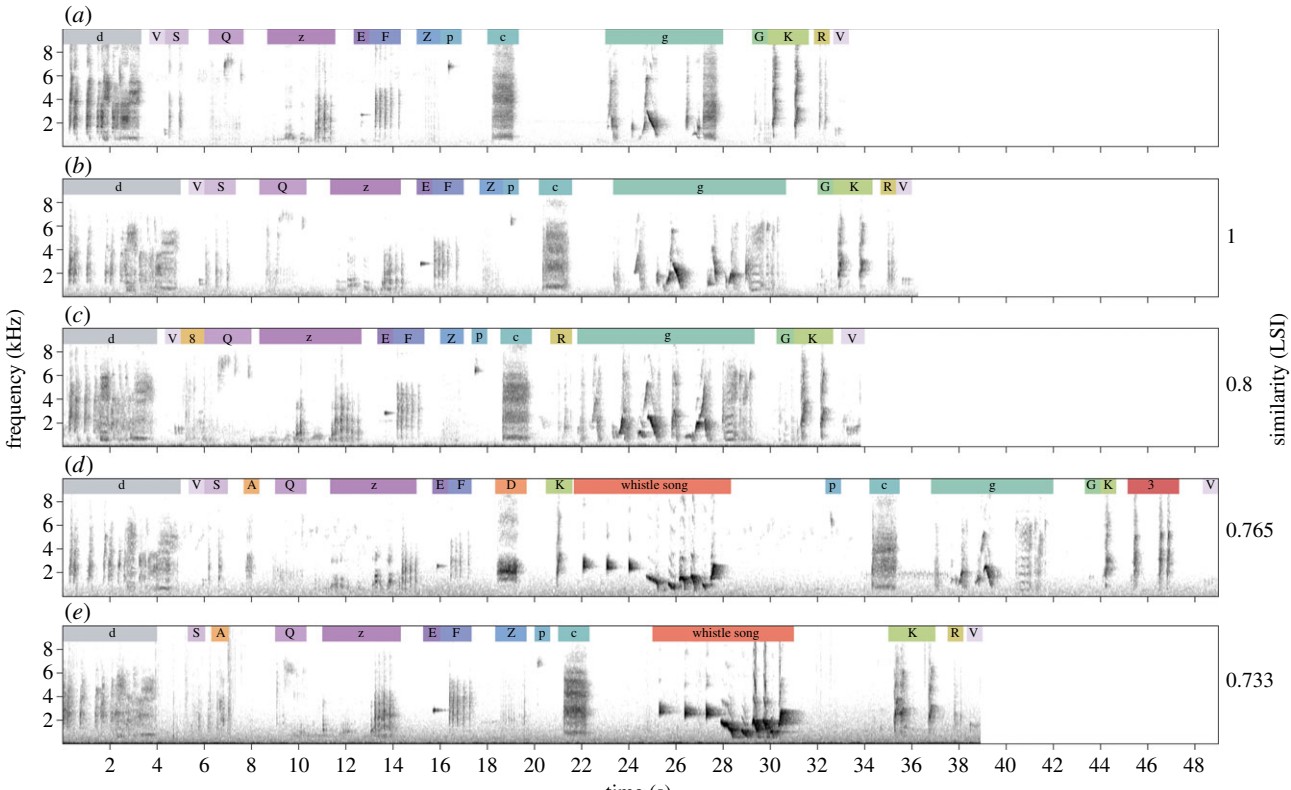

**Figure 1.** Example sequences of mimetic units from five males (*a*–*e*) from the Binna Burra population. Unit categories are: d, satin bowerbird; V, satin bowerbird; S, taps (type 1); Q, wingbeats/white-browed scrubwren; z, laughing kookaburra; E, eastern yellow robin; F, eastern yellow robin; Z, wingbeats; p, green catbird; c, satin bowerbird; g, satin bowerbird; G, crimson rosella; K, crimson rosella; R, taps (type 2); 8, taps (type 3); A, rattle; D, paradise riflebird; 3, Australian king parrot (see electronic supplementary material, table S1). Whistle songs are lyrebird-specific vocalizations and were removed from the LD analysis. Similarities are the LSI between each sequence and the first sequence (*a*). Relevant recordings are provided as electronic supplementary material, files S1–S5.

is presented as a series of multi-element mimetic songs (or 'units') including mimicry of other avian vocalizations and non-vocal sounds such as wingbeats and beak clicking, and makes up at least 90% of vocalizations during a display bout (electronic supplementary material, table S1). We refer to this vocal mimicry as 'recital mimicry', as it is a likely homologue of 'recital mimicry' by superb lyrebirds (*Menura novaehollandiae*) [41,42]. In addition, while experimental testing would be required to determine the function of the recital mimicry, we make the assumption that recital mimicry is 'functionally mimetic' (*sensu* [33]) based on a high acoustic accuracy compared to the original model sounds [39]. In Albert's lyrebirds, mimicked vocalizations appear to be learnt from both conspecific tutors and heterospecific models, although it is unclear how these two mechanisms interact [39]. Sequences of mimetic units in the recital mimicry are typically repeated multiple times without a break, sometimes interrupted at irregular intervals by species-specific whistle songs or introductory notes [40]. Sometimes recital mimicry transitions to a discrete specific-specific song known as 'gronking' [38] (F.B. 2019, personal observation). During the recital mimicry, lyrebirds often hold their ornate tail over their head and sway side-to-side [43] (F.B. 2019, personal observation), but there are no indications that recital mimicry and visual elements are coordinated. Albert's lyrebirds mimic a similar set of model species across the species range, although there are differences between populations in the detailed composition of mimetic repertoires (F.B. 2019, unpublished data).

## (c) Field methods

We recorded 25 adult male Albert's lyrebirds (five males each from the five sites), during the breeding seasons (May–July) of 2018 and 2019. Recordings were made by following individuals as closely as possible without disturbing the birds, usually 15–30 m from the focal bird. Albert's lyrebirds are loud vocalists ([38] (F.B. 2019, personal observation) and so this distance is usually sufficient for high-quality recordings. For six birds, we used recordings from autonomous sound recorders set up 7–10 m from the display platform. Individuals were identified by location [40] and adult male plumage confirmed either in person or from camera trap footage from display platforms. Recordings were made using a hand-held Sennheiser ME 66/ K6 shotgun microphone and a Marantz PMD 661 with a 96 kHz sample rate and 24-bit depth, or a Swift (Terrestrial Passive Acoustic Recording Unit, developed by the Cornell Lab of Ornithology) with a 48 kHz sample rate, 16-bit depth and 33 dB gain.

## (d) Defining units

Approximately 15 min of recital mimicry was selected from the recordings of each male, with effort to choose a continuous bout of recital mimicry with a high signal-to-noise ratio and with minimal species-specific song. Preliminary analysis showed that sequences of mimetic units were up to about one minute long and contained repetition of about 15 different mimetic units, so 15 min was chosen to fill a naïve transition matrix of 15 by 15 units [4]. For 20 out of 25 focal birds, this was continuous mimicry from one recording. For the remaining five, mimicry was taken from 2 different days with as little time between them as possible, with only one bird sampled across 2 years.

Recital mimicry was viewed in Raven Pro 64 bit 1.5 [44]. We identified mimetic units (henceforth 'units') from the spectrogram and by ear, and classified units first to species and then to either vocalization type or, for units comprising mimicry of a non-vocal sound, to acoustic qualities (electronic

supplementary material, appendix S1 provides detail). Most mimetic units contained multiple elements, and some units were comprised of repetitions of short elements (e.g. vocalization 'F' in figure 1). These sequences of repeated elements were considered a single unit as they comprise short phrases and mimic how the vocalization is sung by the model species. Overall, 0.18% (14 of 7859) of mimetic units could not be identified and were classified as 'unknown'. Each unit type was then assigned an alphanumeric code for later analysis (electronic supplementary material, table S1).

## (e) Defining sequences

Preliminary analysis suggested that Albert's lyrebirds organize mimetic units into sequences that are further structured into bouts lasting from several minutes to over an hour. Here, we assume that sequences of mimetic units are Markov chains, where the probability of each unit occurring is determined by a finite number of previous units, although there may be further complex, undetected patterns in sequence organization [45]. As recital mimicry is often presented for an extended period without a break, and whistle songs are sung at inconsistent points within the recital mimicry, the start and endpoints of individual sequences within these bouts were not obvious. To apply the same treatment to all birds and avoid creating artificial differences between populations in sequence length, we split recorded bouts into sequences using the mimetic unit that had the most consistently high occurrence in all birds (satin bowerbird vocalization 'd' in figure 1; electronic supplementary material, appendix S2 and table S3 for details). Song bouts were subsequently split into sequences beginning with this mimetic unit using the *strsplit* function in R. Species-specific whistle songs are temporally discrete within singing bouts [40], and so whistle songs ($n = 314$), and solitary introductory elements normally associated with whistle songs ($n = 14$), were removed from these sequences. Removing whistle songs and introductory elements marginally increased similarities between sequences (electronic supplementary material, table S3), but this is unlikely to affect any conclusions from results. We also removed the first sequence from each individual if it did not begin with the appropriate mimetic unit, and the last sequence to avoid including sequences that are shortened due to recording conditions or external interruptions. This resulted in 448 sequences from the 25 individuals (17.9 mean ± 5.97 s.d. sequences per male, 6593 total mimetic units).

## (f) Sequence similarity

To quantify the variation in mimetic sequences within and between individuals, we calculated the Levenshtein distance (LD) between each possible pair of sequences using the *stringdistmatrix* function from the package 'stringdist' in R [46]. The LD calculates the number of insertions, deletions and substitutions needed to make two strings identical [16,17], and has been shown to be more robust than other methods of sequence analysis [47]. Since sequences were of varying lengths, distances were not directly comparable. Accordingly, we standardized LDs to get a Levenshtein similarity index (LSI) [16,17] using the formula:

$$\text{LSI} = 1 - \left( \frac{\text{LD}}{\max(\text{length(sequence1)}, \text{ length(sequence2)})} \right)$$

This resulted in a matrix of LSIs between all possible pairs of sequences, both within and between individuals (example LSIs shown in figure 1).

Two sequences with a similar repertoire composition will have a higher LSI between them than two sequences of different repertoires regardless of structure, so to test whether sequences were more repeatable than expected given repertoire similarities

we ran a permutation test. We randomized the order of units within each sequence and created another matrix of LSIs, and repeated this 1000 times. For each permutation, we calculated the average LSI for three different groups: within individuals, between individuals from the same population and between individuals from different populations. We then compared the three resulting distributions of permuted averages with the average LSIs for those three groups from the real data using Z-tests in R.

The amount of sequence variation explained by intra-individual variation, differences between individuals from the same population and differences between individuals from different populations was determined with an analysis of molecular variance (AMOVA) using the *amova* function (package 'pegas') in R [48]. AMOVA functions like a nested ANOVA and was developed to measure the amount of genetic variation explained by different hierarchical groups, such as populations within regions [49], and so is ideal for comparing sequence similarities between the three levels examined here. We converted the matrix of LSIs between all sequences to a dissimilarity matrix and ran the AMOVA with individual bird nested within population.

## (g) Geographic analysis

To examine the effect of geographic separation on the degree of sequence sharing, we compared average similarities between pairs of individuals and populations with measures of geographic distance. For distances between populations, we used the length weighted by resistance of least cost paths calculated from a species distribution model (SDM) produced in a previous study [40]. As distances between individuals within populations were often smaller than the resolution of the SDM (range = 95–5927 m, mean = 1709 m), we used geodesic distances between individuals. Average LSI measures were compared with geographic distances with mantel tests in the package 'ade4' [50] using the Monte–Carlo technique and 1000 replicates.

To determine if sequence similarities within populations and differences between populations are driven by model species assemblage, we calculated the likelihood of each model species occurring at each location based on SDMs [51] (electronic supplementary material, appendix S3). Species were deemed likely to occur at a site if the value of the SDM at that site was 0.5 or greater [52].

## (h) Sequence organization

We first determined if the mimetic units are sung with immediate variety by testing whether repeated units occur less than expected by chance. As non-vocal units were given broad classification, we only tested for immediate repetition of vocal mimetic units, which may be objectively identified. Using the same set of sequences as the LSI analysis, we calculated the number of immediately repeated vocal units within all sequences. We then created a permutation test by randomizing the order of units within sequences 1000 times and calculating the number of immediately repeated vocal units across all sequences in each permutation. The real number of immediate repeats was compared with the distribution of expected numbers of repeats using a Z-test.

To determine if the mimetic units are organized to enhance acoustic contrast, we made acoustic measurements of the mimetic units. All recordings used were of sufficient quality to manually classify mimetic units, but not all sequences within these recordings were of sufficient quality for detailed acoustic measurements. Lower quality recording excerpts were discarded *a priori* from the acoustic analysis based on a qualitative assessment of signal strength and background interference. For the remaining 6621 units in high-quality sequences, we drew selection boxes around each mimetic and non-mimetic unit

**Table 1.** Similarity in mimetic sequences. LSI values are observed LSIs between sequences when comparing within individuals, between individuals from the same population and between individuals from different populations, and expected LSIs from randomized sequences from the permutation analysis. LSIs can range from 0 (completely different) to 1 (identical). *p*-values are from the Z-tests comparing observed and expected LSIs.

| group | observed[a] LSI (mean ± s.d.) | expected[b] LSI | *p*-value |
|---|---|---|---|
| within individual | 0.407 ± 0.206 | 0.143 | <0.001 |
| between individuals, same population | 0.356 ± 0.178 | 0.136 | <0.001 |
| between individuals, different populations | 0.193 ± 0.084 | 0.103 | <0.001 |

[a]total range 0.018–1.
[b]total range 0.102–0.146.

**Table 2.** Results of the AMOVA comparing the variation in sequence similarity explained by population and individual bird. Test statistics are $\Phi_{CT}$, within population similarities, compared with all birds; $\Phi_{ST}$, within bird similarities, compared with all birds; $\Phi_{SC}$, within bird similarities, compared with within population similarities. d.f. = degrees of freedom, SSD = sum of squared deviation, MSD = mean squared deviation, VC = variance component.

| source of variation | d.f. | SSD | MSD | VC | % total variance | $\Phi$statistic | *p*-value |
|---|---|---|---|---|---|---|---|
| population | 4 | 39.8 | 9.96 | 0.105 | 31.7 | $\Phi_{CT} = 0.318$ | <0.001 |
| bird | 20 | 13.6 | 0.682 | 0.0274 | 8.31 | $\Phi_{SC} = 0.122$ | <0.001 |
| residual (within bird) | 423 | 83.2 | 0.197 | 0.197 | 59.6 | $\Phi_{ST} = 0.401$ | |
| total | 447 | 137 | 0.306 | | | | |

(including whistle songs) on the spectrogram in Raven using a Hann display type set at Fast Fourier Transform 1024. From these selection boxes, we used Raven to automatically calculate peak frequency, 5% frequency, 95% frequency, 90% bandwidth, aggregate entropy, peak power and 90% duration. While aggregate entropy and power are sensitive to recording conditions, these measurements are valid for comparisons within sequences because any changes within the small timeframe of a single sequence due to recording conditions are likely to be small and random across units.

We assessed the acoustic contrast between consecutive mimetic units within 451 sequences from 24 individuals (mean $18.8 \pm 6.55$ s.d. sequences per individual), including incomplete sequences. Within sequences, we calculated the average difference between each consecutive unit separately for each acoustic variable. To test whether acoustic contrast was higher between consecutive units than expected by chance, we then randomized the order of the units within the sequences and calculated the average acoustic distance for each acoustic variable again. The randomization was repeated 1000 times. The mean acoustic difference between consecutive units across all real sequences was compared with the 1000 corresponding means from the permuted data using a Z-test.

Statistical analyses were run using R v. 4.0.3 [53]. Errors reported are standard deviations.

## 3. Results

### (a) Sequence similarities
Individual Albert's lyrebirds mimicked 11–27 vocalizations from 4 to 11 heterospecifics (including 1–11 vocalization types from each species) and 3–10 other non-vocal sounds. Sequences of mimicry included 3–58 units (mean $15.1 \pm 7.15$) and were on average $38.6 \pm 23.7$ s long, and contained mimicry of $4.80 \pm 1.72$ heterospecifics on average ($n = 448$ sequences from 25 males). Only 35 pairs of sequences out

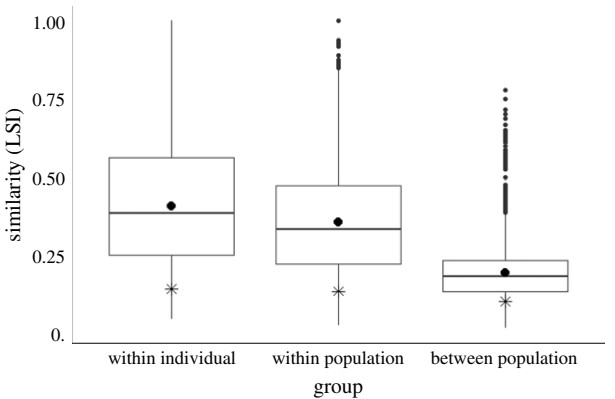

**Figure 2.** The range of all LSIs within individuals, between individuals from the same population and between individuals from different populations. LSIs are different between groups, and significantly higher than expected from random. Mean values indicated with black circles, inclusive median and interquartile ranges indicated with lines, and expected LSI in stars. Maximum possible similarity is LSI = 1.

of a possible 100 128 pairwise comparisons were identical (LSI = 1), and the smallest LSI was 0.0175. The average LSI between all sequence pairs was $0.229 \pm 0.133$.

Mimetic sequences were significantly more similar than pairs of permutated sequences at all levels of comparison (table 1): within individuals ($Z = 257$, $p < 0.001$), between individuals from the same population ($Z = 375$, $p < 0.001$) and between individuals from different populations ($Z = 231$, $p < 0.001$). Population identity explained 31.8% of the variation in sequence structure ($p < 0.001$; table 2 and figure 2), indicating that sequences were more similar within populations than between populations. Sequences were also more similar within individuals than between neighbouring

individuals ($p < 0.001$; table 2 and figure 2), although the differences between individual birds explained only a small amount of the variation (8.44%).

## (b) Geographic differences

Geographic distance did not explain differences in average LSIs between individuals within populations (Binna Burra: $r^2 = 0.132$, $p = 0.149$; Border Ranges: $r^2 = 0.195$, $p = 0.850$; Mt Jerusalem: $r^2 = 0.157$, $p = 0.911$; Tamborine: $r^2 = 0.613$, $p = 0.981$; Goomburra: $r^2 = 0.0220$, $p = 0.301$) or between populations ($r^2 = 0.0698$, $p = 0.631$; see electronic supplementary material, tables S4 and S5 for details).

The habitat suitability for all model species was at least 50% at each site, except for Australian logrunners (*Orthonyx temminckii*) that had a 46.3% chance of occurring at Goomburra (electronic supplementary material, table S6). Most model species are therefore expected to occur at all sites.

## (c) Sequence organization

Visual inspection of the spectrograms revealed that mimetic units were sung with immediate variety. The incidence of repeated vocal units was 16 out of 6593 total mimetic units (0.243% repetition rate), which is significantly lower than the average expected repetition rate across randomized sequences (1.52% expected rate, $p < 0.001$).

When looking at all populations, the contrast between successive units in each acoustic variable was significantly greater than the contrast in the permuted sequences, with the exception of duration, which was significantly smaller within the real sequences (table 3; electronic supplementary material, table S7). These trends were significant in most individual populations, although differences in peak power in the Border Ranges population were significantly lower in the real sequences than the permuted sequences.

# 4. Discussion

We investigated the temporal organization and social transmission of sequences of mimicry sung by male Albert's lyrebirds. Our results suggest that Albert's lyrebirds socially transmit systematically organized sequences of mimetic songs. Furthermore, mimetic units are sung with immediate variety, and mimetic sequences are organized in a manner that enhances acoustic contrast between consecutive units, a sequencing rule that is consistent across the entire species's range and that is likely to enhance receiver perception of repertoire complexity. To our knowledge, this is the first study to quantify the ordering of whole sequences of mimicry, and one of only few studies to show social transmission of higher-order vocal sequences in birds, rather than the smaller constituent parts [11,19].

## (a) Repeatability of sequences within individuals

Our results confirm a long-standing hypothesis that Albert's lyrebirds organize their repertoire of mimetic units into systematically structured sequences [38]. Individual males showed approximately 40% similarity between sequences of vocal mimicry within their own repertoires, which is above random expectation, and places Albert's lyrebird song at the more structured end of a continuum of organization among species. Where similar methods have been used, humpback whales have greater song similarities than Albert's lyrebirds (at least 85% similarity [17]). Differences in methods make it difficult to compare the degree of sequence structure among avian species, but sequence structures range from random song organization (e.g. willow warbler, *Phylloscopus trochilus* [27]) to a varied order (e.g. nightingale, *Luscinia megarhynchos* [12] and Cassin's vireo, *Vireo cassinii* [9]), or consistent order of songs within multi-song 'packages' (e.g. Pallas's warbler, *Phylloscopus proregulus* [54]), to a highly predictable sequence of song types (e.g. fox sparrow, *Passerella iliaca* [15]). Albert's lyrebirds do not appear to order their mimetic units as rigidly as fox sparrows order their songs, although this is unsurprising given their larger repertoires (Albert's lyrebirds: 15–37 mimetic units versus fox sparrows: 2–4 song types).

## (b) Social transmission of sequences

Sequences of vocal mimicry were more similar within males than between neighbouring males and were much more similar within than between populations. Differences between males from the same population were small compared with the differences between populations, implying social transmission of sequences of mimetic units. Alternatively, sequence similarities may be driven by physiological constraints, or local model species assemblages, which in some mimetic species explain geographical differences in repertoires [55,56]. However, all lyrebird populations are probably under similar physiological constraints and able to mimic the same sounds, and all species mimicked had a high chance of occurring at all locations, albeit we could not account for possible differences in the local abundance of model species. Furthermore, sequences in recital mimicry are unlikely to reflect naturally occurring environmental sequences (*sensu* [57]), as recital mimicry includes vocalizations that would be produced by model species in a range of contexts. Another possible explanation is that sequence sharing is a by-product of individuals applying shared acoustic contrast rules to a shared mimetic repertoire. However, given the large mimetic repertoire of Albert's lyrebirds, there are multiple combinations of units that could lead to a high acoustic contrast within the sequence (electronic supplementary material, figure S3), and so population differences in both mimetic repertoires and sequence structure are more likely to reflect social learning. This social transmission of mimicry raises conceptual issues around what constitutes vocal mimicry. It is clear in this species that vocal mimicry can be acquired from both conspecifics and heterospecifics [39], and the role of conspecific tutors in mimicry acquisition further supports a definition of vocal mimicry that does not depend on the mode of acquisition [33].

To our knowledge the sharing of whole, stable sequences of songs between local individuals has rarely been shown in wild populations of birds or mammals. Cultural sharing of a whole acoustic sequence has been shown in humpback whales, where 'songs'—here defined as sequences of multi-element phrases—are shared within populations and progressively transmitted across ocean basins [16,17]. Some avian species have been found to share song transitions or segments of song sequences [11,58,59], and similarities in song-type sequences have been found between neighbouring long-billed marsh wrens (*Cistothorus palustris*) [60]. Our results suggest that during the learning process, Albert's lyrebirds not only perceive vocalizations at the level of individual

**Table 3.** The mean and standard deviation of the mean acoustic differences between consecutive units within sequences (measured) for all populations, and in each individual population, and the corresponding mean acoustic differences measured in the permuted data (expected). Italicized indicates a significant difference between measured and expected values (*** $p < 0.001$, ** $p < 0.01$, * $p < 0.05$). Refer to electronic supplementary material, table S7 for statistical output.

| population | | peak frequency (kHz) | low frequency (kHz) | high frequency (kHz) | 90% bandwidth (kHz) | 90% duration (s) | aggregate entropy (bits) | peak power (dB) |
|---|---|---|---|---|---|---|---|---|
| Binna Burra | measured | 1.70 ± 0.572 | 1.28 ± 0.611 | 1.90 ± 0.377 | 1.96 ± 0.510 | 1.26 ± 0.372 | 1.50 ± 0.392 | 12.9 ± 3.91 |
| | expected | 1.61 | 1.17 | 1.82 | 1.82 | 1.32 | 1.38 | 12.2 |
| | significance | *** | *** | ** | *** | * | *** | *** |
| Border Ranges | measured | 2.01 ± 0.453 | 1.55 ± 0.466 | 1.98 ± 0.392 | 2.09 ± 0.474 | 1.16 ± 0.304 | 1.36 ± 0.423 | 10.3 ± 2.22 |
| | expected | 1.92 | 1.38 | 1.99 | 1.99 | 1.24 | 1.33 | 10.8 |
| | significance | ** | *** | | ** | *** | | ** |
| Goomburra | measured | 1.66 ± 0.616 | 1.59 ± 0.573 | 1.83 ± 0.586 | 1.73 ± 0.705 | 0.884 ± 0.413 | 1.78 ± 0.426 | 12.8 ± 3.85 |
| | expected | 1.53 | 1.45 | 1.75 | 1.65 | 0.943 | 1.63 | 12.9 |
| | significance | *** | *** | ** | ** | ** | *** | |
| Mt Jerusalem | measured | 1.25 ± 0.470 | 0.845 ± 0.302 | 1.57 ± 0.498 | 1.81 ± 0.639 | 1.34 ± 0.401 | 1.28 ± 0.572 | 10.9 ± 3.44 |
| | expected | 1.20 | 0.781 | 1.60 | 1.82 | 1.41 | 1.26 | 11.2 |
| | significance | * | *** | ** | ** | * | *** | |
| Tamborine | measured | 1.84 ± 0.489 | 1.67 ± 0.717 | 1.95 ± 0.448 | 1.67 ± 0.332 | 1.24 ± 0.445 | 1.30 ± 0.329 | 14.5 ± 3.72 |
| | expected | 1.67 | 1.45 | 1.74 | 1.58 | 1.59 | 1.22 | 13.0 |
| | significance | *** | *** | *** | * | *** | ** | *** |
| all populations | measured | 1.67 ± 0.588 | 1.38 ± 0.615 | 1.83 ± 0.498 | 1.85 ± 0.594 | 1.15 ± 0.426 | 1.49 ± 0.483 | 12.2 ± 3.77 |
| | expected | 1.57 | 1.25 | 1.78 | 1.78 | 1.25 | 1.40 | 12.1 |
| | significance | *** | *** | *** | *** | *** | *** | * |

units, where mimetic units may be learnt at least in part from heterospecific models [39], but may also perceive the temporal relationships between units when learning from conspecifics. Our results thus highlight the importance of considering vocalizations at the levels of both songs and song sequences.

Signals that are socially transmitted may change with geographic distance [61]. Sequences were more similar within than between populations, implying an effect of geographic distance on sequence similarities. However, similarities between males within populations were not correlated with geographic distance, and similarities between populations did not vary with geographic distance between populations. The lack of geographic pattern between populations contrasts with patterns found in the whistle song of Albert's lyrebirds, where differences in whole whistle songs, as well as in the body and final element of the whistle song, are correlated with geographic distance between populations [40]. However, population differences in the mimetic introductory notes of the whistle song did not correlate with geographic distance. Unlike conspecific song, both these introductory notes and the recital mimicry reported in the present study could be affected by subtle differences in the local availability of model species, potentially confounding the relationship between geographic and acoustic distance at the inter-population level. Additionally, the lack of correlation between sequence similarity and geographic distance between populations may reflect limited cultural transfer between populations of a highly sedentary species (sensu [62]). Within populations, sequence differences between individuals may be too small to detect a signal of geographic distance given the variation found within individual birds, or the variation in geographic distances within populations may be insufficient to test this relationship.

## (c) Sequence function

Across the species range, we found two patterns that may be species-wide rules to organization. First, Albert's lyrebirds present their mimetic units with immediate variety, with unit types over-dispersed within sequences. Second, Albert's lyrebirds enhance acoustic contrast by juxtaposing acoustically dissimilar units more than expected by chance. The recital mimicry of lyrebirds is suspected to function to attract mates [41,63,64], and the combination of singing with immediate variety and increasing acoustic contrast between successive units strongly implies that mimetic sequences have been selected to maximize perceived repertoire variation or complexity. All acoustic measurements showed high contrast between consecutive units, with the exception of duration, which was similar between consecutive units. Differences in temporal measures may be more difficult for females to perceive, or may be less stimulating than differences in other spectral measures. Many other species sing with eventual variety, in which song types are repeated many times before changing, which may allow individuals to showcase a high degree of accuracy in their songs [65]. Conversely, Albert's lyrebirds appear to favour diversity by singing with immediate variety, although accuracy is still likely important. Further, superb lyrebirds appear to abridge mimetic units by removing repetitions of certain elements within imitated multi-element sounds, while maintaining the original element order and number of element types [64], suggesting that both extant lyrebird species use several strategies to enhance the perception of mimetic

repertoire complexity, while maintaining the complexity of individual mimetic units. Using contrast to increase the efficacy of signals has been shown in visual signals [25] but rarely in acoustic signals [10,28,31]. In other bird species, an increased contrast in acoustic structure between consecutive songs can increase the aggressive response of a male receiver [28,31], or has been hypothesized to function in mate attraction [10]. Surprisingly, northern mockingbirds, another species known for vocal mimicry, differ from Albert's lyrebirds in singing with gradual acoustic changes between mimetic units, although the function of this is unclear [35]. We recommend that future studies on sequence function consider the role of acoustic contrast in sequence structure and function.

The immediate variety of and acoustic contrast among the mimetic units in Albert's lyrebirds has implications for the drivers of mimicry in both lyrebirds and other mimicking species. The structure of mimicry at both unit and sequence levels in both lyrebird species implies that females select for both mimetic diversity and accuracy. Despite being highly accurate and versatile vocal mimics [39], Albert's lyrebirds do not use their full mimetic repertoire during the recital mimicry, given their ability to mimic other heterospecifics during sub-song (e.g. pied currawong, yellow-tailed black cockatoo, sulfur-crested cockatoo; F.B. 2019, personal observation). Instead, Albert's lyrebirds may exhibit their mimetic abilities by mimicking accurately a limited number of particularly acoustically arresting heterospecific vocalizations, something that requires a study on model choice to confirm. Furthermore, higher-order organization of mimetic song has received little attention across mimicking species, with the exception of studies on northern mockingbirds [34,35], and a comparison of sequence organization across mimicking species would greatly help in understanding both the drivers of model choice and the function of vocal mimetic behaviours.

## 5. Conclusion

While many studies on animal acoustic communication focus on individual units of communication, animals across several taxa communicate in higher-order temporal sequences [13–16]. Social transmission of such sequences occurs in humpback whales and a limited number of oscine passerines [11,16], but may be found in other taxa exhibiting vocal learning, such as parrots, hummingbirds, bats and other cetaceans. In addition, reasons for specific higher-order sequences are largely unexplained. Higher-order sequences may be neglected due to historical difficulties obtaining long recordings and the complexities of making detailed comparisons [4,47]. We suggest that advances in technology allowing autonomous long-term acoustic recordings combined with the methods here, including use of a permuted version of the LSI and comparing acoustic attributes between units, provide a useful approach that could be applied to other species, as well as in other fields involving sequences. The acoustic structure and social transmission of sequences of mimicry of Albert's lyrebirds, and other organisms, suggests that higher-order sequences are an important source of cultural diversity worthy of attention.

**Ethics.** All work for this study was approved by the Western Sydney University Animal Care and Ethics Committee (no. A12077) and data collected under Scientific Research Permits from the NSW Parks and Wildlife Service (no. SL101351) and the QLD Parks and Wildlife Service (no. WITK18768218).

Data accessibility. The data and R code used in the analysis are available from the Dryad Digital Repository: https://doi.org/10.5061/dryad.05qfttf45 [66]. R v. 4.0.3 is required for the analysis.

Authors' contributions. F.B.: conceptualization, data curation, formal analysis, investigation, methodology writing—original draft and writing—review and editing; A.H.D.: conceptualization, funding acquisition, methodology, resources, supervision and writing—review and editing; R.D.M.: conceptualization, methodology, supervision and writing—review and editing; J.A.W.: conceptualization, funding acquisition, investigation, methodology, resources, supervision and writing—review and editing.

All authors gave final approval for publication and agreed to be held accountable for the work performed therein.

Competing interests. We declare we have no competing interests.

Funding. This research was supported by an Australian Government Research Training Program scholarship through Western Sydney University (F.B.), Birdlife Northern NSW (F.B.), the Cornell Lab of Ornithology Rose Postdoctoral Fellowship Program (A.H.D.), a University of Wollongong VC Postdoctoral Fellowship (A.H.D.), the Hawkesbury Institute for the Environment (J.A.W.) and an NSF grant no. 1730791 (A.H.D. and J.A.W.).

Acknowledgements. Special thanks to Hannah Mirando and Tristan Herwood for assistance in data collection. We are grateful to David Putland for early conversations on Albert's lyrebird behaviour and analysis methods, and to Annabel Dorrestein for invaluable help in creating the R functions used in the analysis. We also thank two anonymous reviewers and Arik Kershenbaum for their valuable suggestions.

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
