## [Peer Review File · Proceedings of the Royal Society B: Biological Sciences]

Review History

RSPB-2021-1859.R0 (Original submission)

Review form: Reviewer 1

Recommendation

Major revision is needed (please make suggestions in comments)

Scientific importance: Is the manuscript an original and important contribution to its field?

Excellent

General interest: Is the paper of sufficient general interest?

Good

Quality of the paper: Is the overall quality of the paper suitable?

Good

Is the length of the paper justified?

Yes

Should the paper be seen by a specialist statistical reviewer?

No

Do you have any concerns about statistical analyses in this paper? If so, please specify them explicitly in your report.

No

It is a condition of publication that authors make their supporting data, code and materials available - either as supplementary material or hosted in an external repository. Please rate, if applicable, the supporting data on the following criteria.

Is it accessible?

Yes

Is it clear?

Yes

Is it adequate?

Yes

Do you have any ethical concerns with this paper?

No

Comments to the Author

Backhouse et al. have performed an impressive study of Albert's lyrebirds that reveals a clear temporal pattern of animal communication that is daunting to demonstrate empirically, though easily conceived on a theoretical level. I agree strongly with their insight that "while many studies treat songs as individual units of analysis, the higher-level organization of songs into sequences may too encode biologically relevant information, beyond the individual units" (lines 41-43). Indeed, for human communication it is laughable to consider the approach that communication structure could be fully understood by a narrow focus on just words (while ignoring phrases, sentences, paragraphs, etc.), yet that simplistic approach describes most of how researchers think about animal communication (with a trickle of attempts to study larger sequences, such as studies on humpback whale songs). Sequences of animal communication are probably ignored because of the empirical difficulty of the research, e.g., obtaining long sequences from several individuals, coding all song types objectively, making detailed bioacoustics comparisons. I commend Backhouse et al. for overcoming these challenges, and I look forward to applying their approaches in my own research.

Although I find their study impressive overall, I have a few methodological concerns, plus other general comments, followed by several specific comments:

General comment #1 – definition of 'immediate variety': For decades researchers have batted around the terms 'immediate variety' and 'eventual variety' to contrast the ways in which songbirds present their song repertoires. Backhouse et al. want to convince readers that Albert's lyrebirds present with immediate variety, so they employ a precise definition that involves complex measurements and technical statistical analysis. The details of this definition need clarification, and to be honest, I'm not sure 'immediate variety' needs definition; the approach taken by Backhouse et al. seems more like a solution in search of a problem. Looking at Figure 1, it is obvious that these lyrebirds switch between mimetic units every few seconds, so I suspect nearly all readers will already be convinced the lyrebirds sing 'immediate variety.' I recommend dropping the empirical demonstration of immediate variety and simply referring readers to Fig 1. This would shorten the manuscript and improve readability. If the authors feel compelled to keep the technical analysis, here are a few concerns to resolve:

Backhouse et al. define immediate variety based on 'repetition of units' (line 238), but it is unclear to me how they identified repetition of units. For example, mimetic unit F shown on Fig 1 always contains an element that is repeated, yet each of the five panels of Fig 1 shows just one F rather

than several Fs. Why is F considered to be the cluster of elements rather than the elements themselves? Given that mimetic units can apparently include repeated elements, what would it even look like if F were repeated by a bird? (I illustrate this problem with mimetic unit F, but similar examples exist for many other mimetic units.) Answering these concerns will help readers to understand the assertion that only 29 of 7859 mimetic units were repeated (line 305).

I also worry there might be a flaw in the underlying assumption of lines 237-239, in which the authors calculate 'the highest chance of the repetition of units as one divided by the total number of unit types.' If I'm not misunderstanding, this calculation seems to rely on a null assumption of the random ordering of units - an assumption that is unlikely to be met in most systems of animal communication, given that most (all?) species produce some units much more frequently than other units. Once production frequencies are taken into account, the null expectations for repeating units would be lower, which would make it less likely for the authors to reach the conclusion of immediate variety.

In sum, defining 'immediate variety' in a precise way might provide slight marginal gains, but it also provides non-trivial marginal costs because of theoretical concerns about the underlying assumptions and technical concerns over how (and whether) to apply their methods to other species. I was also surprised when the authors asserted that another prominent vocal mimic, the northern mockingbird, uses eventual rather than immediate variety (lines 403-405) - an assertion not shared by naturalists and researchers who are familiar with mockingbird song. Based on what I have read and observed about mockingbirds, I would argue their songs use immediate variety, similar to what is shown in Fig 1 for Albert's lyrebird.

General concern #2 - LSI methods: Given our primitive understanding of sequences in animal communication, the study of Backhouse et al. is likely to be replicated in other species, so it is important to explain their methods clearly. Based on the work of Kershenbaum and others, Backhouse et al. recommend the Levenshtein Similarity Index (LSI) as the best way to quantify sequence similarity. I'm ok with that judgment, but I had a difficult time seeing the connections between their descriptions of how to make LSI measurements (see lines 193-204) and Figure 1. For example, when I compare Panel a and Panel e of Fig 1, I calculate a Levenshtein distance of 4 (delete V, add A, delete g, delete G), and an LSI of $1 - 4/\max(15,13) = 1-4/15 = 0.733$, not 0.765. I suspect I am missing something, but I recommend the authors illustrate their LSI calculations using an example from Fig 1.

General concern #3 - more natural history needed, especially early in Methods: I'm always a big fan of including more natural history in these types of studies because of the critical role natural history plays in constraining study design, methods and interpretation of results. At the same time, I know journals like Proc B are going to want a streamlined study, so I will prioritize my list of what natural history is lacking based on the relative importance of items within this list.

Most importantly, to what extent are 'extended singing bouts' (line 140) a multimodal display? The remainder of your study simplifies these displays down to a sequence of mimetic units, so I want to understand what information might have been left out of your analysis. Is it possible that dance components or other visual signals also form stereotyped sequences that might be integrated with the stereotyped sequences of mimetic units?

When/how could social learning take place within Albert's lyrebirds? Social learning of vocal sequences requires listening and social reinforcement, but apparently adult males live an average of 1.7km apart from other adult males (line 228) - much too far for neighbors to listen to each other. I'm sure there are contexts in which social learning could take place, but these contexts are not currently described in the paper. It would therefore be great to add a few sentences to the Discussion, e.g., right after line 357.

More background is needed on the noise experienced by singing Albert's lyrebirds (and therefore encountered by the field researchers). Noise can severely disrupt any study that measures

acoustic similarity and is often prominent when researchers record birds from a great distance (15-30m from focal bird – see lines 152-154). There was apparently enough noise to force you to discard an unspecified number of sequences from your acoustic analysis (line 243), and it also prevented you from measuring aggregate entropy and power (lines 249-250). Any evidence that distortions from noise might have altered your results?

What do we know about model selection in Albert's lyrebird? If lyrebirds are trying to highlight immediate acoustic variety, then they might sample models with acoustically-bizarre songs, while not sampling model species with acoustically bland songs. I would share examples, but I'm unfortunately not familiar enough with Australian species. (If a rigorous study of model selection has not been performed for Albert's lyrebird, then I think you should recommend such a study – perhaps around lines 421-424.)

Do female Albert's lyrebirds sing? If so, do they mimic?

In line 140 'extended' could mean a minute up to several hours. Can you be any more precise?

Please clarify the percentage of acoustic units within an extended singing bout that are mimetic vs nonmimetic. Fig 1 looks like at least 90% mimetic, with the only nonmimetic signals being whistle songs and gronking. Is that correct?

Do whistle songs and gronking occur as repertoires? Your analysis currently focuses on just the mimetic units, and one of your main conclusions is that sequential mimetic units have high acoustic contrast. If whistle songs and/or gronking consists of repertoires, then these signals might potentially contribute to the presentation of acoustic variety.

General concern #4 – Duration results need more attention: The data of Table 3 (and Table S7) show convincingly that temporally-adjacent mimetic units are acoustically-dissimilar with regard to most of their acoustic measurements. This is an important result, particularly because it highlights what is likely a key difference in the songs of Albert's lyrebird and the northern mockingbird (see Roeske et al. 2021). That said, I find it interesting that the duration results are of comparable magnitude, but in the opposite direction. This result is mostly ignored in the Discussion. I am wondering what might be driving this result (e.g., respiratory constraints? – see the work of Franz Goller and others). I also wonder whether the authors might need to soften their conclusion, given that four of their seven measures are all frequency measurements (likely correlated, plus frequency measurements are often distorted by variation in noise – see lines 242-252 combined with the research of Sue Ann Zollinger), and given that many potentially useful acoustic measurements, such as spectral continuity and frequency excursion, were apparently not measured.

General concern #5 – suggestion of future studies: Although the discussion was well-written, I felt it was a little too focused on just Albert's lyrebirds. Given the ground-breaking approach of Backhouse et al., I recommend broadened the taxonomic focus within the Discussion. For example, is it possible to connect more to studies of song/call sequences performed in humpback whales, hyraxes, marsh wrens, etc.? I also recommend broadening their disciplinary focus. For example, the methods described in lines 179-184 address a challenge like the problem faced by computational biologists when they categorize DNA sequences within a chromosome, which consists of millions of nucleotides. (If you want to go even broader, your challenge is also like that faced in business and media analytics, in which researchers look for patterns in massive streams of data.) What methodological strategies have been used successfully by computational biologists that might be applied to the study of animal communication sequences? For example, might there be a song sequence in Albert's lyrebird communication that is analogous to the 'promoter' sequences that identify the start of genes?

One final recommendation for a study to recommend within the discussion: I think the time is ripe for a comparative study of song structure in prominent vocal mimics, such as European

starlings (Goller), icterine warblers (Riegert), northern mockingbirds (Gammon), greater racket-tailed drongos (Goodale), marsh warblers (Dowsett-Lemaire), satin bowerbirds (Coleman), superb lyrebirds (Dalziell), and perhaps other mimicking species. In the past few decades, researchers in vocal mimicry have often blazed their own trails, without giving much attention to replicating the methods used in other studies. A comparative study would help to crystalize similarities and differences between mimicking species.

Specific concerns:

Line 38: Given the general readership of Proc B, I recommend you start the manuscript with a sentence or two about the importance of higher-order structure in human communication.

Line 39: revise 'avian communication' to 'avian acoustic communication'

Line 62-63: I recommend rewording 'singing with little to no immediate repetition of song or unit types.' Otherwise some readers might misinterpret your meaning, because Fig 1 shows plenty of repeated syllable units within a single 'mimetic unit' (see General concern #1).

Lines 131-135: Consider adding a map of the five study sites as a supplementary figure, or referring to a map in another manuscript (perhaps Putland et al. 2006?).

Line 151: In superb lyrebirds adult males look and sing differently than sub-adult males (see Zann & Dunstan 2008). Is this also true for Albert's lyrebirds? If so, how did you account for these developmental differences in the design of your study?

Line 164: Does 'where possible' mean 19 of 25? If so, please add the numbers?

Line 179: I recommend adding a phrase, so that it reads "...structured into bouts that can last for several minutes." ('Bout' is defined differently by researchers studying other songbirds, so the added phrase clarifies how you define 'bout'.)

Line 183: I'm splitting hairs here, but the satin bowerbird vocalization on the left side of Fig 1 doesn't look to me like a 'long' buzz. Rather, it looks like a short buzz that gets repeated many times. (It's probably fine with me if you ignore this comment.)

Line 191-192: Do the parenthetical numbers refer to mean +/- one standard deviation?

Line 216: If I understand correctly, an AMOVA functions like a nested ANOVA. If this is true, I recommend you add a short phrase here to clarify, because many (most?) birdsong readers will be unfamiliar with AMOVA. I initially thought it was a misprint.

Line 260-261: Forgive me for my statistical ignorance, but is 'standard deviation of the mean' the same as 'standard error'?

Line 268: I think you meant 'an,' not 'on'

Line 286: More context needed for the three Greek term listed in Table 2. I assume these three numbers somehow connect to the information presented in Table 2, but I don't yet see the connection.

Line 288-292: What comparison(s) do you want readers to make for Figure 2? If I understand correctly, the main story of Figure 2 is that each of the three LSI averages is considerably higher than the three expected LSI averages. If my thinking is correct, then I recommend adding three expectation lines and inserting a 'take home message' into the figure heading. Right now, the figure looks like you might be trying to highlight that the 1st and 2nd column means are

significantly higher than the 3rd column mean; I'm pretty sure that's not the point you want readers to make.

Line 323: Would the 'duet codes' studied by David Logue and others be another example of social transmission of vocal sequences in birds? (In brief, Logue found that the antiphonal 'FMFMFM' duets sung by Male and Female wrens normally use codes, e.g., female song #12 always alternates with male song #26 and no other male song type, female song #4 alternates with male song # 18 and no other male song type.) I'm guessing you would not consider these as vocal sequences and that you could leave the manuscript intact, but I thought I'd raise the question, just in case.

Line 345-357: Great reasoning; this paragraph felt triumphant.

Lines 385-390: These two rules seem well-supported by your data, but I don't see how the two rules by themselves would lead to the similarity of sequences across all the geographic range.

Lines 397-398: What exactly does it mean to 'abridge mimetic units by removing internal repetition within imitated sounds'? Does this mean that if a superb lyrebird wants to imitate a multi-syllable model vocalization, that it only one of the syllables within the model vocalization while dropping the other syllables?

Line 439: I think you mean 'Special' not 'Species'

Review form: Reviewer 2

Recommendation

Major revision is needed (please make suggestions in comments)

Scientific importance: Is the manuscript an original and important contribution to its field?

Good

General interest: Is the paper of sufficient general interest?

Good

Quality of the paper: Is the overall quality of the paper suitable?

Acceptable

Is the length of the paper justified?

Yes

Should the paper be seen by a specialist statistical reviewer?

No

Do you have any concerns about statistical analyses in this paper? If so, please specify them explicitly in your report.

No

It is a condition of publication that authors make their supporting data, code and materials available - either as supplementary material or hosted in an external repository. Please rate, if applicable, the supporting data on the following criteria.

Is it accessible?

Yes

Is it clear?

Yes

Is it adequate?

Yes

Do you have any ethical concerns with this paper?

No

Comments to the Author

This manuscript looks at the structure of lyrebird songs, a species that mimics other species and includes some of these mimetic songs into their own song. The authors find that lyrebirds produce stereotyped song sequences, where the also the mimetic elements occur in stereotyped sequences. Given that these sequences are similar within populations, the authors argue that this reflects social transmission. Moreover, these sequences maximize the contrast between the elements.

The use of a species that mimics other species provides a nice opportunity to study song structure. Clearly, this is trickier in species that to not mimic other species, but it is still possible. I think that there have been more studies on song that did look into the song sequences, and how this is socially acquired, at least what I could see when scanning for literature. Sure, this must be studied in zebra finches too?

Although lyrebird song includes mimetic elements, the structure of the song seems to be acquired socially. I therefore wonder how these mimetic elements are learned ontogenetically. Are they learned from other lyrebirds, or picked up from the species that are mimicked? Given that the species which are mimicked do not occur everywhere in the study population, I speculate that they are also socially transmitted, along with the structure of the song. Thus, although these elements are originally copied from other species, if they are learned mostly from conspecifics, then it's strictly speaking not mimicry anymore.

I assume the authors do not have any genetic and/or life-history information about the individuals used in the study? These data would be helpful to support the argument of the authors.

Please see in the enclosed PDF for more comments.

Decision letter (RSPB-2021-1859.R0)

01-Oct-2021

Dear Miss Backhouse:

I am writing to inform you that your manuscript RSPB-2021-1859 entitled "Sequences of vocal mimicry performed by male Albert's lyrebirds are socially transmitted and enhance acoustic contrast" has, in its current form, been rejected for publication in Proceedings B.

This action has been taken on the advice of referees, who have recommended that substantial revisions are necessary. With this in mind we would be happy to consider a resubmission, provided the comments of the referees are fully addressed. However please note that this is not a provisional acceptance.

The resubmission will be treated as a new manuscript. However, we will approach the same reviewers if they are available and it is deemed appropriate to do so by the Editor. Please note that resubmissions must be submitted within six months of the date of this email. In exceptional

circumstances, extensions may be possible if agreed with the Editorial Office. Manuscripts submitted after this date will be automatically rejected.

Sincerely,
Dr Robert Barton
mailto:proceedingsb@royalsociety.org

Associate Editor

Comments to Author:

We have now received two reviews of your manuscript. Both the reviewers and I found much to appreciate about your study. It is an impressive study of vocal sequence structure in lyrebirds – a powerful but challenging system to work in. There are interesting novel data that are presented clearly. However, the manuscript has several flaws that prevent me from accepting it in its current form. The main problem is that several interesting features (mimicry, sequence structure, song function) are introduced but none of them are explored with sufficient depth. For mimicry, there appear to be interesting combinations of between species mimicry and within species copying but we are not given enough information to understand how this works. For example, the sequences appear to be socially transmitted but are the individual mimetic elements also socially transmitted? If so, does it make sense to call them mimetic elements? For sequence structure, there is interesting evidence of copying at the sequence level but this is not well situated in the current literature on sequence learning in birds or other species. For this part of the analysis, it does not appear important that the song elements originated in other species which makes the sequence learning less novel but also easier to compare to the current literature (which needs more careful review). For the functional part, it is interesting and suggestive that adjacent elements tend to be acoustically dissimilar but there is not enough here to make strong conclusions about how they combine elements and how combinations are responded to. It seems beyond the scope of a single manuscript to cover each of these topics with the necessary depth. However, it may be possible to narrow the focus of the paper to provide a more thorough and cohesive analysis. To me, the most exciting part is the song-level matching across individuals despite such diverse vocal elements (shown in Figure 1). I think this could be the focus of the paper and compared to other species where, at least in birds, song sequence similarity has been seen more often in species with limited repertoires of notes. With greater focus there would be more opportunity to explore where the variation in sequence structure lies and how it comes about. In general, more could be done to tease apart two distinct causes for deviations from random combinations; preferences for certain combinations/patterns of sounds vs. copying from neighbors. Do they learn blocks of units? Are some combinations of units particularly widespread? How does this relate to their apparent preference for juxtaposing contrasting units? If the paper can be focused and the central theme strengthened, I would be willing to consider a resubmission that would go out to review (hopefully to the same reviewers). Note that a more

focused manuscript would not preclude raising some other issues (e.g., mimicry, female preference) in the discussion.

Reviewer(s)' Comments to Author:

Referee: 1

Comments to the Author(s)

Backhouse et al. have performed an impressive study of Albert's lyrebirds that reveals a clear temporal pattern of animal communication that is daunting to demonstrate empirically, though easily conceived on a theoretical level. I agree strongly with their insight that "while many studies treat songs as individual units of analysis, the higher-level organization of songs into sequences may too encode biologically relevant information, beyond the individual units" (lines 41-43). Indeed, for human communication it is laughable to consider the approach that communication structure could be fully understood by a narrow focus on just words (while ignoring phrases, sentences, paragraphs, etc.), yet that simplistic approach describes most of how researchers think about animal communication (with a trickle of attempts to study larger sequences, such as studies on humpback whale songs). Sequences of animal communication are probably ignored because of the empirical difficulty of the research, e.g., obtaining long sequences from several individuals, coding all song types objectively, making detailed bioacoustics comparisons. I commend Backhouse et al. for overcoming these challenges, and I look forward to applying their approaches in my own research.

Although I find their study impressive overall, I have a few methodological concerns, plus other general comments, followed by several specific comments:

General comment #1 – definition of 'immediate variety': For decades researchers have batted around the terms 'immediate variety' and 'eventual variety' to contrast the ways in which songbirds present their song repertoires. Backhouse et al. want to convince readers that Albert's lyrebirds present with immediate variety, so they employ a precise definition that involves complex measurements and technical statistical analysis. The details of this definition need clarification, and to be honest, I'm not sure 'immediate variety' needs definition; the approach taken by Backhouse et al. seems more like a solution in search of a problem. Looking at Figure 1, it is obvious that these lyrebirds switch between mimetic units every few seconds, so I suspect nearly all readers will already be convinced the lyrebirds sing 'immediate variety.' I recommend dropping the empirical demonstration of immediate variety and simply referring readers to Fig 1. This would shorten the manuscript and improve readability. If the authors feel compelled to keep the technical analysis, here are a few concerns to resolve:

Backhouse et al. define immediate variety based on 'repetition of units' (line 238), but it is unclear to me how they identified repetition of units. For example, mimetic unit F shown on Fig 1 always contains an element that is repeated, yet each of the five panels of Fig 1 shows just one F rather than several Fs. Why is F considered to be the cluster of elements rather than the elements themselves? Given that mimetic units can apparently include repeated elements, what would it even look like if F were repeated by a bird? (I illustrate this problem with mimetic unit F, but similar examples exist for many other mimetic units.) Answering these concerns will help readers to understand the assertion that only 29 of 7859 mimetic units were repeated (line 305).

I also worry there might be a flaw in the underlying assumption of lines 237-239, in which the authors calculate 'the highest chance of the repetition of units as one divided by the total number of unit types.' If I'm not misunderstanding, this calculation seems to rely on a null assumption of the random ordering of units – an assumption that is unlikely to be met in most systems of animal communication, given that most (all?) species produce some units much more frequently than other units. Once production frequencies are taken into account, the null expectations for repeating units would be lower, which would make it less likely for the authors to reach the conclusion of immediate variety.

In sum, defining ‘immediate variety’ in a precise way might provide slight marginal gains, but it also provides non-trivial marginal costs because of theoretical concerns about the underlying assumptions and technical concerns over how (and whether) to apply their methods to other species. I was also surprised when the authors asserted that another prominent vocal mimic, the northern mockingbird, uses eventual rather than immediate variety (lines 403-405) – an assertion not shared by naturalists and researchers who are familiar with mockingbird song. Based on what I have read and observed about mockingbirds, I would argue their songs use immediate variety, similar to what is shown in Fig 1 for Albert’s lyrebird.

General concern #2 – LSI methods: Given our primitive understanding of sequences in animal communication, the study of Backhouse et al. is likely to be replicated in other species, so it is important to explain their methods clearly. Based on the work of Kershenbaum and others, Backhouse et al. recommend the Levenshtein Similarity Index (LSI) as the best way to quantify sequence similarity. I’m ok with that judgment, but I had a difficult time seeing the connections between their descriptions of how to make LSI measurements (see lines 193-204) and Figure 1. For example, when I compare Panel a and Panel e of Fig 1, I calculate a Levenshtein distance of 4 (delete V, add A, delete g, delete G), and an LSI of $1 - 4/\max(15,13) = 1-4/15 = 0.733$, not 0.765. I suspect I am missing something, but I recommend the authors illustrate their LSI calculations using an example from Fig 1.

General concern #3 – more natural history needed, especially early in Methods: I’m always a big fan of including more natural history in these types of studies because of the critical role natural history plays in constraining study design, methods and interpretation of results. At the same time, I know journals like Proc B are going to want a streamlined study, so I will prioritize my list of what natural history is lacking based on the relative importance of items within this list.

Most importantly, to what extent are ‘extended singing bouts’ (line 140) a multimodal display? The remainder of your study simplifies these displays down to a sequence of mimetic units, so I want to understand what information might have been left out of your analysis. Is it possible that dance components or other visual signals also form stereotyped sequences that might be integrated with the stereotyped sequences of mimetic units?

When/how could social learning take place within Albert’s lyrebirds? Social learning of vocal sequences requires listening and social reinforcement, but apparently adult males live an average of 1.7km apart from other adult males (line 228) – much too far for neighbors to listen to each other. I’m sure there are contexts in which social learning could take place, but these contexts are not currently described in the paper. It would therefore be great to add a few sentences to the Discussion, e.g., right after line 357.

More background is needed on the noise experienced by singing Albert’s lyrebirds (and therefore encountered by the field researchers). Noise can severely disrupt any study that measures acoustic similarity and is often prominent when researchers record birds from a great distance (15-30m from focal bird – see lines 152-154). There was apparently enough noise to force you to discard an unspecified number of sequences from your acoustic analysis (line 243), and it also prevented you from measuring aggregate entropy and power (lines 249-250). Any evidence that distortions from noise might have altered your results?

What do we know about model selection in Albert’s lyrebird? If lyrebirds are trying to highlight immediate acoustic variety, then they might sample models with acoustically-bizarre songs, while not sampling model species with acoustically bland songs. I would share examples, but I’m unfortunately not familiar enough with Australian species. (If a rigorous study of model selection has not been performed for Albert’s lyrebird, then I think you should recommend such a study – perhaps around lines 421-424.)

Do female Albert’s lyrebirds sing? If so, do they mimic?

In line 140 ‘extended’ could mean a minute up to several hours. Can you be any more precise?

Please clarify the percentage of acoustic units within an extended singing bout that are mimetic vs nonmimetic. Fig 1 looks like at least 90% mimetic, with the only nonmimetic signals being whistle songs and gronking. Is that correct?

Do whistle songs and gronking occur as repertoires? Your analysis currently focuses on just the mimetic units, and one of your main conclusions is that sequential mimetic units have high acoustic contrast. If whistle songs and/or gronking consists of repertoires, then these signals might potentially contribute to the presentation of acoustic variety.

General concern #4 – Duration results need more attention: The data of Table 3 (and Table S7) show convincingly that temporally-adjacent mimetic units are acoustically-dissimilar with regard to most of their acoustic measurements. This is an important result, particularly because it highlights what is likely a key difference in the songs of Albert’s lyrebird and the northern mockingbird (see Roeske et al. 2021). That said, I find it interesting that the duration results are of comparable magnitude, but in the opposite direction. This result is mostly ignored in the Discussion. I am wondering what might be driving this result (e.g., respiratory constraints? – see the work of Franz Goller and others). I also wonder whether the authors might need to soften their conclusion, given that four of their seven measures are all frequency measurements (likely correlated, plus frequency measurements are often distorted by variation in noise – see lines 242-252 combined with the research of Sue Ann Zollinger), and given that many potentially useful acoustic measurements, such as spectral continuity and frequency excursion, were apparently not measured.

General concern #5 – suggestion of future studies: Although the discussion was well-written, I felt it was a little too focused on just Albert’s lyrebirds. Given the ground-breaking approach of Backhouse et al., I recommend broadened the taxonomic focus within the Discussion. For example, is it possible to connect more to studies of song/call sequences performed in humpback whales, hyraxes, marsh wrens, etc.? I also recommend broadening their disciplinary focus. For example, the methods described in lines 179-184 address a challenge like the problem faced by computational biologists when they categorize DNA sequences within a chromosome, which consists of millions of nucleotides. (If you want to go even broader, your challenge is also like that faced in business and media analytics, in which researchers look for patterns in massive streams of data.) What methodological strategies have been used successfully by computational biologists that might be applied to the study of animal communication sequences? For example, might there be a song sequence in Albert’s lyrebird communication that is analogous to the ‘promoter’ sequences that identify the start of genes?

One final recommendation for a study to recommend within the discussion: I think the time is ripe for a comparative study of song structure in prominent vocal mimics, such as European starlings (Goller), icterine warblers (Riegert), northern mockingbirds (Gammon), greater racket-tailed drongos (Goodale), marsh warblers (Dowsett-Lemaire), satin bowerbirds (Coleman), superb lyrebirds (Dalziell), and perhaps other mimicking species. In the past few decades, researchers in vocal mimicry have often blazed their own trails, without giving much attention to replicating the methods used in other studies. A comparative study would help to crystalize similarities and differences between mimicking species.

Specific concerns:

Line 38: Given the general readership of Proc B, I recommend you start the manuscript with a sentence or two about the importance of higher-order structure in human communication.

Line 39: revise ‘avian communication’ to ‘avian acoustic communication’

Line 62-63: I recommend rewording 'singing with little to no immediate repetition of song or unit types.' Otherwise some readers might misinterpret your meaning, because Fig 1 shows plenty of repeated syllable units within a single 'mimetic unit' (see General concern #1).

Lines 131-135: Consider adding a map of the five study sites as a supplementary figure, or referring to a map in another manuscript (perhaps Putland et al. 2006?).

Line 151: In superb lyrebirds adult males look and sing differently than sub-adult males (see Zann & Dunstan 2008). Is this also true for Albert's lyrebirds? If so, how did you account for these developmental differences in the design of your study?

Line 164: Does 'where possible' mean 19 of 25? If so, please add the numbers?

Line 179: I recommend adding a phrase, so that it reads "...structured into bouts that can last for several minutes." ('Bout' is defined differently by researchers studying other songbirds, so the added phrase clarifies how you define 'bout.')

Line 183: I'm splitting hairs here, but the satin bowerbird vocalization on the left side of Fig 1 doesn't look to me like a 'long' buzz. Rather, it looks like a short buzz that gets repeated many times. (It's probably fine with me if you ignore this comment.)

Line 191-192: Do the parenthetical numbers refer to mean +/- one standard deviation?

Line 216: If I understand correctly, an AMOVA functions like a nested ANOVA. If this is true, I recommend you add a short phrase here to clarify, because many (most?) birdsong readers will be unfamiliar with AMOVA. I initially thought it was a misprint.

Line 260-261: Forgive me for my statistical ignorance, but is 'standard deviation of the mean' the same as 'standard error'?

Line 268: I think you meant 'an,' not 'on'

Line 286: More context needed for the three Greek term listed in Table 2. I assume these three numbers somehow connect to the information presented in Table 2, but I don't yet see the connection.

Line 288-292: What comparison(s) do you want readers to make for Figure 2? If I understand correctly, the main story of Figure 2 is that each of the three LSI averages is considerably higher than the three expected LSI averages. If my thinking is correct, then I recommend adding three expectation lines and inserting a 'take home message' into the figure heading. Right now, the figure looks like you might be trying to highlight that the 1st and 2nd column means are significantly higher than the 3rd column mean; I'm pretty sure that's not the point you want readers to make.

Line 323: Would the 'duet codes' studied by David Logue and others be another example of social transmission of vocal sequences in birds? (In brief, Logue found that the antiphonal 'FMFMFM' duets sung by Male and Female wrens normally use codes, e.g., female song #12 always alternates with male song #26 and no other male song type, female song #4 alternates with male song # 18 and no other male song type.) I'm guessing you would not consider these as vocal sequences and that you could leave the manuscript intact, but I thought I'd raise the question, just in case.

Line 345-357: Great reasoning; this paragraph felt triumphant.

Lines 385-390: These two rules seem well-supported by your data, but I don't see how the two rules by themselves would lead to the similarity of sequences across all the geographic range.

Lines 397-398: What exactly does it mean to 'abridge mimetic units by removing internal repetition within imitated sounds?' Does this mean that if a superb lyrebird wants to imitate a multi-syllable model vocalization, that it only one of the syllables within the model vocalization while dropping the other syllables?

Line 439: I think you mean 'Special' not 'Species'

Referee: 2

Comments to the Author(s)

This manuscript looks at the structure of lyrebird songs, a species that mimics other species and includes some of these mimetic songs into their own song. The authors find that lyrebirds produce stereotyped song sequences, where the also the mimetic elements occur in stereotyped sequences. Given that these sequences are similar within populations, the authors argue that this reflects social transmission. Moreover, these sequences maximize the contrast between the elements.

The use of a species that mimics other species provides a nice opportunity to study song structure. Clearly, this is trickier in species that to not mimic other species, but it is still possible. I think that there have been more studies on song that did look into the song sequences, and how this is socially acquired, at least what I could see when scanning for literature. Sure, this must be studied in zebra finches too?

Although lyrebird song includes mimetic elements, the structure of the song seems to be acquired socially. I therefore wonder how these mimetic elements are learned ontogenetically. Are they learned from other lyrebirds, or picked up from the species that are mimicked? Given that the species which are mimicked do not occur everywhere in the study population, I speculate that they are also socially transmitted, along with the structure of the song. Thus, although these elements are originally copied from other species, if they are learned mostly from conspecifics, then it's strictly speaking not mimicry anymore.

I assume the authors do not have any genetic and/or life-history information about the individuals used in the study? These data would we helpful to support the argument of the authors.

Please see in the enclosed PDF for more comments.

Author's Response to Decision Letter for (RSPB-2021-1859.R0)

See Appendix A.

RSPB-2021-2498.R0

Review form: Reviewer 1

Recommendation

Accept with minor revision (please list in comments)

Scientific importance: Is the manuscript an original and important contribution to its field?

Good

General interest: Is the paper of sufficient general interest?

Acceptable

Quality of the paper: Is the overall quality of the paper suitable?

Excellent

Is the length of the paper justified?

Yes

Should the paper be seen by a specialist statistical reviewer?

No

Do you have any concerns about statistical analyses in this paper? If so, please specify them explicitly in your report.

No

It is a condition of publication that authors make their supporting data, code and materials available - either as supplementary material or hosted in an external repository. Please rate, if applicable, the supporting data on the following criteria.

Is it accessible?

Yes

Is it clear?

Yes

Is it adequate?

Yes

Do you have any ethical concerns with this paper?

No

Comments to the Author

I was Reviewer #1 on the original submission. After spending several hours looking through the comments of the editor, me, and R2, and the detailed responses of Backhouse et al., I can only conclude the authors did a thorough job of addressing our criticisms. I still maintain a couple of important disagreements (see below), but these come down to differences of opinion between reasonable researchers about which assumptions should be foundational. As a fellow researcher in vocal mimicry, I applaud the high quality of their scientific study and believe it represents an important contribution in the study of higher order sequences of animal vocalizations.

I am particularly intrigued by two aspects of the study. First, the authors demonstrate similarity of higher order sequences within and between individuals – an important result based on LSI and permutation test methods that could be applied to any sequence of animal vocalizations. The implication of the result for Albert's lyrebird is particularly interesting, i.e., that lyrebirds acquire their mimetic songs at least partially through social transmission from conspecifics rather than heterospecifics. This implication is not intuitive for researchers in vocal mimicry and represents an interesting puzzle that should stimulate future research. The second aspect of Backhouse et al.'s study that intrigues me is their result showing an acoustic contrast between subsequent song units, suggesting that recital mimicry functions to highlight to females the acoustic diversity of the signaler.

I have only one minor criticism, and it relates to the accuracy of the LSI measure in panel d of Fig. 1. (Thank you for fixing the LSI in panel e.) Comparing d to a, and after ignoring the whistle song, I see four changes (add A, Z□D, add K, and R□3), with $1 - (4/17)$ not equal to 0.8. Please help me to understand your math.

I now have two general comments, but as I mentioned above, these comments come down to differences of opinion, not criticisms of the quality of the study.

General comment #1 – my original concerns over immediate variety remain unresolved: I now see some value in how the authors rigorously quantified immediate variety, but the marginal benefit to their study still seems low. I'm also not convinced their methods would apply well beyond their specific study. Although Backhouse et al. adequately addressed my earlier criticisms, their justification in lines 175-177 seems to apply well only to sequences of vocal mimicry. If the sequence comes from a non-mimicking species (i.e., most species), or if the sequence comes from the non-mimetic song of a vocal mimic species, then it is impossible to assess whether something like vocalization F in Fig 1 would represent a single unit vs repetition of a single unit (element in this case). Even for mimicking species, the methods of Backhouse et al. might provide strange results for 'immediate' variety, e.g., if the mimic faithfully copies a model species that repeats an element for 20s or more. Given the authors already have a convincing result on acoustic contrast, I recommend dropping what seems like unnecessary data on immediate variety, but as I mentioned above, I do see some value in their approach and I am willing to yield to their opinion.

General comment #2 – definition of vocal mimicry: The authors rely heavily on the rigorous definition of vocal mimicry recommended in Dalziell et al. (2015), which focuses on function. The authors are correct that this definition is widely employed and consistent with theoretical explanations for other types of animal mimicry, such as Batesian mimicry. I nevertheless argue for a little caution about adopting a 'function-only' definition wholesale, given how little we know about vocal mimicry and given that vocal mimicry (as defined by Dalziell et al. 2015) occurs only in birds that learn their songs. Another way to define vocal mimicry focuses on development rather than function, and a developmental definition was apparently favored by at least Reviewer #2 and the editor (and myself for that matter). Another reason I urge caution is that the study of Backhouse et al. does not provide any direct evidence for function in the higher order song sequences they study. Instead, all their data focus on the behavior of senders. Without measuring the behavioral responses of receivers listening to the mimicry, the authors cannot clear the high bar set by the definition used by Dalziell et al. 2015. Overall, I don't have a problem with Backhouse et al. using this definition in their paper, but it might be worth adding one or two caveats to the manuscript.

Review form: Reviewer 3

Recommendation

Accept with minor revision (please list in comments)

Scientific importance: Is the manuscript an original and important contribution to its field?

Good

General interest: Is the paper of sufficient general interest?

Good

Quality of the paper: Is the overall quality of the paper suitable?

Excellent

Is the length of the paper justified?

Yes

Should the paper be seen by a specialist statistical reviewer?

No

Do you have any concerns about statistical analyses in this paper? If so, please specify them explicitly in your report.

No

It is a condition of publication that authors make their supporting data, code and materials available - either as supplementary material or hosted in an external repository. Please rate, if applicable, the supporting data on the following criteria.

Is it accessible?

Yes

Is it clear?

Yes

Is it adequate?

Yes

Do you have any ethical concerns with this paper?

No

Comments to the Author

This is a well-designed and well executed study, which has done a nice job of analysing song sequences in a rigorous and effective way. I'm joining as a reviewer after the first revision, so I'm afraid that I may be suggesting some changes that would have been more appropriate on first submission, but I can reassure the authors that I believe the study is well worth publishing, and hopefully the additional analysis should not be particularly onerous. Also, I am waiving anonymity, as I am recommending that the authors consult a couple of my own papers.

I am very pleased to see the extensive use of exact tests to measure the significance of the findings. If only more studies would make use of this! As not all researchers are familiar with randomisation tests, it is always worth restating the hypothesis (e.g. on line 249: that repeated units occur less likely than would be expected by chance). Table 1 could do with giving the p values from the Z test.

One little quibble that I have is the use of "higher order" as a description for the patterns being studied. While I think that the term is technically accurate (first order relationships being higher order than simple repertoire diversity), it might be confused with the kinds of hierarchical structures seen in, for instance, humpback whale song. Although the authors do a good job of looking at transition probabilities, I myself would not call this "higher order". I'm not overly fussed about this, but I think it's worth making the point that the authors are not, for instance, looking at repetition of motifs.

In addition, where discussing the repetitions (line 319), it's worth mentioning the implicit assumption that the sequences are simple Markov chains, i.e. that the probability of a unit occurring can be predicted by the preceding unit(s). If there are indeed higher order structures, then this assumption is unlikely to be met. However, the observation that the repetition rate is lower than expected is an important one, as most animal vocal sequences appear to have greater repetition (see Kershenbaum et al. (2014). *Animal vocal sequences: not the Markov chains we thought they were*. *Proceedings of the Royal Society B: Biological Sciences*).

For the geographic analysis, I would refer the authors to my paper: Kershenbaum et al. (2012). *Syntactic structure and geographical dialects in the songs of male rock hyraxes*. *Proceedings of the Royal Society B: Biological Sciences*. This would be a relevant comparison, because the songs are group territorial songs, and therefore perhaps more likely to be perceived consistently by neighbours.

I'm a little worried about the acoustic similarity analysis. Were the metrics standardised (e.g. to zero mean and unit SD, or alternatively to a range 0-1)? If not, it is hard to calculate mean acoustic difference, if the metrics have different orders of magnitude.

I think that the conclusion (line 332) that there is social transmission is a weak one. There are other explanations that should be discussed. In particular, I think that the observations could be explained by a combination of the acoustic contrast rules, together with a difference in repertoires between the different populations. I would say that it would be a very useful (and easy) additional test to see whether there is significant difference in the repertoire distributions or not. Particularly as you state (line 430) that the repertoire is limited, it seems essential to tell the readers whether the repertoires are the same (or similar) between populations. Additionally, I think the conclusion on line 377, that the birds perceive entire sequences, is also not well supported, as it hasn't been tested explicitly. Perhaps it would be possible to make this speculative claim if hierarchical structures were found, but the similarity between overall sequences could be explained by simpler, short-term rules. I think it would be good to qualify this conclusion. It's still a very interesting result!

A few minor points:

Line 164: How were the songs selected?

Line 184: Delete "In order".

Line 186: It is explained that sequences begin with an instance of the most common unit. Could you provide some rationale for this, as it is not immediately obvious why this should be the criterion?

Line 191: You remove whistle songs. Does this mean that units on either side of the whistle song are treated as being adjacent? This could cause problems for the sequence analysis, as they are not genuinely adjacent. Some discussion of this would be sufficient

Signed review:

Arik Kershenbaum

Decision letter (RSPB-2021-2498.R0)

04-Jan-2022

Dear Miss Backhouse:

Your manuscript has now been peer reviewed and the reviews have been assessed by an Associate Editor. The reviewers' comments (not including confidential comments to the Editor) and the comments from the Associate Editor are included at the end of this email for your reference. As you will see, the reviewers and the Editors have raised some concerns with your manuscript and we would like to invite you to revise your manuscript to address them.

When submitting your revision please upload a file under "Response to Referees" in the "File Upload" section. This should document, point by point, how you have responded to the reviewers' and Editors' comments, and the adjustments you have made to the manuscript. We

require a copy of the manuscript with revisions made since the previous version marked as 'tracked changes' to be included in the 'response to referees' document.

Research ethics:

Use of animals and field studies:

It is a condition of publication that you make available the data and research materials supporting the results in the article (<https://royalsociety.org/journals/authors/author-guidelines/#data>). Datasets should be deposited in an appropriate publicly available repository and details of the associated accession number, link or DOI to the datasets must be included in the Data Accessibility section of the article (<https://royalsociety.org/journals/ethics-policies/data-sharing-mining/>). Reference(s) to datasets should also be included in the reference list of the article with DOIs (where available).

If you wish to submit your data to Dryad (<http://datadryad.org/>) and have not already done so you can submit your data via this link [http://datadryad.org/submit?journalID=RSPB&manu=\(Document not available\)](http://datadryad.org/submit?journalID=RSPB&manu=(Document%20not%20available)), which will take you to your unique entry in the Dryad repository.

Online supplementary material will also carry the title and description provided during submission, so please ensure these are accurate and informative. Note that the Royal Society will not edit or typeset supplementary material and it will be hosted as provided. Please ensure that

the supplementary material includes the paper details (authors, title, journal name, article DOI). Your article DOI will be 10.1098/rspb.[paper ID in form xxxx.xxxx e.g. 10.1098/rspb.2016.0049].

Please submit a copy of your revised paper within three weeks. If we do not hear from you within this time your manuscript will be rejected. If you are unable to meet this deadline please let us know as soon as possible, as we may be able to grant a short extension.

Best wishes,
Dr Robert Barton
mailto:proceedingsb@royalsociety.org

Associate Editor Board Member

Comments to Author:

We have now received two reviews of your revised manuscript. Both the reviewers and I find the manuscript much improved. However, the reviewers point to areas for further improvement. These largely involve softening or clarifying some of the conclusions and further acknowledging alternative explanations and interpretations. Most notably, Reviewer 1 points out that the evidence for social transmission of sequences is weak given that similarity above chance might be the product of shared repertoires and transition tendencies (a shared tendency to increase contrast between units). Reviewer 1 suggests testing this directly but, short of that, this weakness should be acknowledged. Reviewer 2 has concerns about the immediate variety analysis particularly as it relates to repeated elements (like vocalization F in figure 1). Perhaps it would help to indicate that the duration and timing of repetitions in such units also support treating them as a single element (in addition to the mimicry-based criteria given in lines 175-177). Reviewer 2 also has concerns about social transmission and the definition of mimicry. I share this concern. While I understand the utility of a broad definition of mimicry that does not imply a mechanism (as in Batesian mimics), with vocal mimicry there is in implication that there is direct copying between species (as stated in the definition of vocal mimicry on line 76-77). This definition really does focus on the mechanism of transmission (rather than the function). This seems to be a larger issue with the use of the term 'mimicry' in this context which is beyond the scope of this manuscript. However, the ambiguity with the meaning of vocal mimicry in relation to the results seems worth mentioning in the Discussion. Both reviewers make additional suggestions for clarification and improvement.

Reviewer(s)' Comments to Author:

Referee: 3

Comments to the Author(s).

This is a well-designed and well executed study, which has done a nice job of analysing song sequences in a rigorous and effective way. I'm joining as a reviewer after the first revision, so I'm afraid that I may be suggesting some changes that would have been more appropriate on first submission, but I can reassure the authors that I believe the study is well worth publishing, and hopefully the additional analysis should not be particularly onerous. Also, I am waiving anonymity, as I am recommending that the authors consult a couple of my own papers. I am very pleased to see the extensive use of exact tests to measure the significance of the findings. If only more studies would make use of this! As not all researchers are familiar with randomisation tests, it is always worth restating the hypothesis (e.g. on line 249: that repeated units occur less likely than would be expected by chance). Table 1 could do with giving the p values from the Z test.

One little quibble that I have is the use of "higher order" as a description for the patterns being studied. While I think that the term is technically accurate (first order relationships being higher order than simple repertoire diversity), it might be confused with the kinds of hierarchical structures seen in, for instance, humpback whale song. Although the authors do a good job of looking at transition probabilities, I myself would not call this "higher order". I'm not overly

fussed about this, but I think it's worth making the point that the authors are not, for instance, looking at repetition of motifs.

In addition, where discussing the repetitions (line 319), it's worth mentioning the implicit assumption that the sequences are simple Markov chains, i.e. that the probability of a unit occurring can be predicted by the preceding unit(s). If there are indeed higher order structures, then this assumption is unlikely to be met. However, the observation that the repetition rate is lower than expected is an important one, as most animal vocal sequences appear to have greater repetition (see Kershenbaum et al. (2014). *Animal vocal sequences: not the Markov chains we thought they were*. *Proceedings of the Royal Society B: Biological Sciences*).

For the geographic analysis, I would refer the authors to my paper: Kershenbaum et al. (2012). *Syntactic structure and geographical dialects in the songs of male rock hyraxes*. *Proceedings of the Royal Society B: Biological Sciences*. This would be a relevant comparison, because the songs are group territorial songs, and therefore perhaps more likely to be perceived consistently by neighbours.

I'm a little worried about the acoustic similarity analysis. Were the metrics standardised (e.g. to zero mean and unit SD, or alternatively to a range 0-1)? If not, it is hard to calculate mean acoustic difference, if the metrics have different orders of magnitude.

I think that the conclusion (line 332) that there is social transmission is a weak one. There are other explanations that should be discussed. In particular, I think that the observations could be explained by a combination of the acoustic contrast rules, together with a difference in repertoires between the different populations. I would say that it would be a very useful (and easy) additional test to see whether there is significant difference in the repertoire distributions or not. Particularly as you state (line 430) that the repertoire is limited, it seems essential to tell the readers whether the repertoires are the same (or similar) between populations.

Additionally, I think the conclusion on line 377, that the birds perceive entire sequences, is also not well supported, as it hasn't been tested explicitly. Perhaps it would be possible to make this speculative claim if hierarchical structures were found, but the similarity between overall sequences could be explained by simpler, short-term rules. I think it would be good to qualify this conclusion. It's still a very interesting result!

A few minor points:

Line 164: How were the songs selected?

Line 184: Delete "In order".

Line 186: It is explained that sequences begin with an instance of the most common unit. Could you provide some rationale for this, as it is not immediately obvious why this should be the criterion?

Line 191: You remove whistle songs. Does this mean that units on either side of the whistle song are treated as being adjacent? This could cause problems for the sequence analysis, as they are not genuinely adjacent. Some discussion of this would be sufficient

Signed review:

Arik Kershenbaum

Referee: 1

Comments to the Author(s).

I was Reviewer #1 on the original submission. After spending several hours looking through the comments of the editor, me, and R2, and the detailed responses of Backhouse et al., I can only conclude the authors did a thorough job of addressing our criticisms. I still maintain a couple of important disagreements (see below), but these come down to differences of opinion between reasonable researchers about which assumptions should be foundational. As a fellow researcher in vocal mimicry, I applaud the high quality of their scientific study and believe it represents an important contribution in the study of higher order sequences of animal vocalizations.

I am particularly intrigued by two aspects of the study. First, the authors demonstrate similarity of higher order sequences within and between individuals – an important result based on LSI and

permutation test methods that could be applied to any sequence of animal vocalizations. The implication of the result for Albert's lyrebird is particularly interesting, i.e., that lyrebirds acquire their mimetic songs at least partially through social transmission from conspecifics rather than heterospecifics. This implication is not intuitive for researchers in vocal mimicry and represents an interesting puzzle that should stimulate future research. The second aspect of Backhouse et al.'s study that intrigues me is their result showing an acoustic contrast between subsequent song units, suggesting that recital mimicry functions to highlight to females the acoustic diversity of the signaler.

I have only one minor criticism, and it relates to the accuracy of the LSI measure in panel d of Fig. 1. (Thank you for fixing the LSI in panel e.) Comparing d to a, and after ignoring the whistle song, I see four changes (add A, Z□D, add K, and R□3), with $1 - (4/17)$ not equal to 0.8. Please help me to understand your math.

I now have two general comments, but as I mentioned above, these comments come down to differences of opinion, not criticisms of the quality of the study.

General comment #1 – my original concerns over immediate variety remain unresolved: I now see some value in how the authors rigorously quantified immediate variety, but the marginal benefit to their study still seems low. I'm also not convinced their methods would apply well beyond their specific study. Although Backhouse et al. adequately addressed my earlier criticisms, their justification in lines 175-177 seems to apply well only to sequences of vocal mimicry. If the sequence comes from a non-mimicking species (i.e., most species), or if the sequence comes from the non-mimetic song of a vocal mimic species, then it is impossible to assess whether something like vocalization F in Fig 1 would represent a single unit vs repetition of a single unit (element in this case). Even for mimicking species, the methods of Backhouse et al. might provide strange results for 'immediate' variety, e.g., if the mimic faithfully copies a model species that repeats an element for 20s or more. Given the authors already have a convincing result on acoustic contrast, I recommend dropping what seems like unnecessary data on immediate variety, but as I mentioned above, I do see some value in their approach and I am willing to yield to their opinion.

General comment #2 – definition of vocal mimicry: The authors rely heavily on the rigorous definition of vocal mimicry recommended in Dalziell et al. (2015), which focuses on function. The authors are correct that this definition is widely employed and consistent with theoretical explanations for other types of animal mimicry, such as Batesian mimicry. I nevertheless argue for a little caution about adopting a 'function-only' definition wholesale, given how little we know about vocal mimicry and given that vocal mimicry (as defined by Dalziell et al. 2015) occurs only in birds that learn their songs. Another way to define vocal mimicry focuses on development rather than function, and a developmental definition was apparently favored by at least Reviewer #2 and the editor (and myself for that matter). Another reason I urge caution is that the study of Backhouse et al. does not provide any direct evidence for function in the higher order song sequences they study. Instead, all their data focus on the behavior of senders. Without measuring the behavioral responses of receivers listening to the mimicry, the authors cannot clear the high bar set by the definition used by Dalziell et al. 2015. Overall, I don't have a problem with Backhouse et al. using this definition in their paper, but it might be worth adding one or two caveats to the manuscript.

Author's Response to Decision Letter for (RSPB-2021-2498.R0)

See Appendix B.

RSPB-2021-2498.R1

Review form: Reviewer 1

Recommendation

Accept as is

Scientific importance: Is the manuscript an original and important contribution to its field?

Good

General interest: Is the paper of sufficient general interest?

Acceptable

Quality of the paper: Is the overall quality of the paper suitable?

Good

Is the length of the paper justified?

Yes

Should the paper be seen by a specialist statistical reviewer?

No

Do you have any concerns about statistical analyses in this paper? If so, please specify them explicitly in your report.

No

It is a condition of publication that authors make their supporting data, code and materials available - either as supplementary material or hosted in an external repository. Please rate, if applicable, the supporting data on the following criteria.

Is it accessible?

Yes

Is it clear?

Yes

Is it adequate?

Yes

Do you have any ethical concerns with this paper?

No

Comments to the Author

I'm generally happy with the changes the authors have made. Just a few (typographic) minor points:

Line 177: Should be "composed" of...

Line 244: This reference is of a different format.

Line 298: Worth mentioning the test type (AMOVA) together with the p value.

Review form: Reviewer 3

Recommendation

Accept as is

Scientific importance: Is the manuscript an original and important contribution to its field?

Good

General interest: Is the paper of sufficient general interest?

Good

Quality of the paper: Is the overall quality of the paper suitable?

Excellent

Is the length of the paper justified?

Yes

Should the paper be seen by a specialist statistical reviewer?

No

Do you have any concerns about statistical analyses in this paper? If so, please specify them explicitly in your report.

No

It is a condition of publication that authors make their supporting data, code and materials available - either as supplementary material or hosted in an external repository. Please rate, if applicable, the supporting data on the following criteria.

Is it accessible?

Yes

Is it clear?

Yes

Is it adequate?

Yes

Do you have any ethical concerns with this paper?

No

Comments to the Author

I was Reviewer #1 on the original submission, and Reviewer #2 on the second submission. As I stated in my second review, I didn't have significant criticisms of the quality of the study, and I still don't. They addressed my minor concern over the mathematical calculations in Fig 1d, and their responses to the concerns of Reviewer #3 also seem adequate. I agree with the authors that the alternative hypothesis proposed by Reviewer #3 regarding non-social transmission seems implausible. I also agree with the authors that conducting Rev3's recommended analysis would take a lot of journal space without adding much benefit to the manuscript.

I appreciate the responses of the authors to my two general concerns in my second review, even though they still don't convince me. High acoustic fidelity could be evidence of selection for functional mimicry as they assume, but in studies of mimicry the relationship between fidelity and functionality is not always perfect. Consider, for example, the visual mimicry in the eggs of brood parasites. Some eggs are accepted by hosts (i.e., functionally mimetic) even though untrained human undergraduates can distinguish the eggs of parasite vs host (i.e., low fidelity). Further, some eggs are rejected by hosts (i.e., functionally non-mimetic) even though expert human birders can hardly distinguish the eggs of parasite vs host (i.e., high fidelity). I also find it compelling that vocal mimicry occurs only in songbird species that learn their vocalizations, which makes me hesitant to exclude development from the definition of vocal mimicry. Even if mimicry is learned from conspecifics, (which is such a cool concept), the mimicry still can be

traced backwards in time to an imitation of a heterospecific. ...But as I asserted in my second review, these issues come down to differences of opinion, not criticisms of the quality of the study. And I agree fully with the authors that the proper definition of vocal mimicry is beyond the scope of their paper.

Decision letter (RSPB-2021-2498.R1)

09-Feb-2022

Dear Miss Backhouse

I am pleased to inform you that your manuscript entitled "Higher-order sequences of vocal mimicry performed by male Albert's lyrebirds are socially transmitted and enhance acoustic contrast" has been accepted for publication in Proceedings B.

Data Accessibility section

Open Access

Paper charges

Sincerely,
Dr Robert Barton
Editor, Proceedings B
mailto: proceedingsb@royalsociety.org

Associate Editor:
Board Member: 1
Comments to Author:
(There are no comments.)

Board Member: 2
Comments to Author:
(There are no comments.)

Appendix A

Revision comments - Fiona Backhouse, Anastasia H. Dalziell, Robert D. Magrath, Justin A. Welbergen

Manuscript ID RSPB-2021-1859 – Higher-order sequences of vocal mimicry performed by male Albert’s lyrebirds are socially transmitted and enhance acoustic contrast

Please note that all comments from the reviewers are in normal font, and our replies to the comments are in **boldface**. All ‘new’ line numbers refer to “ALBsequences_MAINTTEXT_R1_clean”. All changes to the text are marked as tracked changes in “ALBsequences_MAINTTEXT_R1_trackedchanges”.

Associate Editor Comments to Author:

We have now received two reviews of your manuscript. Both the reviewers and I found much to appreciate about your study. It is an impressive study of vocal sequence structure in lyrebirds—a powerful but challenging system to work in. There are interesting novel data that are presented clearly. However, the manuscript has several flaws that prevent me from accepting it in its current form. The main problem is that several interesting features (mimicry, sequence structure, song function) are introduced but none of them are explored with sufficient depth. For mimicry, there appear to be interesting combinations of between species mimicry and within species copying but we are not given enough information to understand how this works. For example, the sequences appear to be socially transmitted but are the individual mimetic elements also socially transmitted? If so, does it make sense to call them mimetic elements? For sequence structure, there is interesting evidence of copying at the sequence level but this is not well situated in the current literature on sequence learning in birds or other species. For this part of the analysis, it does not appear important that the song elements originated in other species which makes the sequence learning less novel but also easier to compare to the current literature (which needs more careful review). For the functional part, it is interesting and suggestive that adjacent elements tend to be acoustically dissimilar but there is not enough here to make strong conclusions about how they combine elements and how combinations are responded to. It seems beyond the scope of a single manuscript to cover each of these topics with the necessary depth. However, it may be possible to narrow the focus of the paper to provide a more thorough and cohesive analysis. To me, the most exciting part is the song-level matching across individuals despite such diverse vocal elements (shown in Figure 1). I think this could be the focus of the paper and compared to other species where, at least in birds, song sequence similarity has been seen more often in species with limited repertoires of notes. With greater focus there would be more opportunity to explore where the variation in sequence structure lies and how it comes about. In general, more could be done to tease apart two distinct causes for deviations from random combinations; preferences for certain combinations/patterns of sounds vs. copying from neighbors. Do they learn blocks of units? Are some combinations of units particularly widespread? How does this relate to their apparent preference for juxtaposing contrasting units? If the paper can be focused and the central theme strengthened, I would be willing to consider a resubmission that would go out to review (hopefully to the same reviewers). Note that a more focused manuscript would not preclude raising some other issues (e.g., mimicry, female preference) in the discussion.

[We thank the editor for their praise of the study as well as helpful suggestions. There are a few issues highlighted by the editor that we address here, as well as in more detail where these issues are also raised by the reviewers.]

The editor has suggested narrowing the focus of the paper, and has included some very interesting ideas that would be great for future studies. However, we believe that we should retain the current focus of the manuscript on both the function and acquisition of higher-order sequences. Our paper is unusual in focussing on higher-order signal organisation, in contrast to most studies that focus on the smaller constituent parts (e.g. song elements or short, multi-element songs). In addition, our novel finding that unit structure enhances acoustic contrast further strengthens the central argument of the paper that higher-order acoustic structure is a key component of non-human animal acoustic communication, and is thus essential to this paper. The results of the acoustic contrast analysis further show that the order of the mimicry has a specific function, which places this study another step ahead of studies that typically only quantify the order of smaller constituent parts (e.g., song elements or short, multi-element songs). Exploring the mechanisms by which variation in sequence structure arises, or how specific units are combined, as suggested by the editor, would result in a different paper, but one which would only be possible after the advances we report here.

We have strengthened the central theme by making the following changes:

- The opening sentence now draws a comparison with human language and music, in order to emphasise the importance of considering higher-level song organisation (lines 38 - 41)
- We have clarified throughout the MS that we are focussing on higher-order song organisation, to distinguish this study from those considering sequences of smaller units such as elements or syllables within songs
- We have strengthened a paragraph in the discussion explaining how acoustic contrast within the sequence may be an important component of vocal mimicry (lines 426 - 440)
- We have changed the final paragraph of the discussion to emphasise the novelty and importance of investigating the function and acquisition of higher-order sequence structures (lines 442 - 455)

The Editor and Reviewer 2 have also highlighted an interesting discussion point on whether mimetic elements can still be considered mimicry if they are learnt from other lyrebirds. We argue that the mimetic elements sung by Albert's lyrebirds should be considered mimicry.

First, to not refer to mimetic elements as mimicry would represent an abrupt departure from conventional theory, where the definition of mimicry is not dependent on the mode of acquisition (genetic or learnt) nor the sensory modality (acoustic, visual, etc.) through which it is expressed (e.g. Wickler 2013, Dalziell and Welbergen 2016b, Jamie 2017, de Jager and Anderson 2019). In our paper we define vocal mimicry in the same way as mimicry in other modalities (Dalziell et al. 2015), in line with contemporary practice.

Second, while there is evidence that mimetic elements are socially transmitted, lyrebirds also learn from the models. For example, they copy local model dialects (Putland et al. 2006).

The editor further suggested more comparison with the current literature on vocal sequences. As pointed out by R1, few researchers have studied the higher-order organisation of songs, and within these studies it is rare to investigate social transmission or functional reasons for sequence order. There are no previous studies of the organisation of mimetic sequences, with the exception of studies on northern

mockingbirds that investigate transitions between consecutive units (Gammon and Altizer 2011, Roeske et al. 2021), but not overall sequence structure. Reviewer 2 provided some interesting references focussed on the organisation of syllables or phrases within songs (Cody et al. 2016, Zsebók et al. 2020, Paul et al. 2021), the organisation of song types (Benitez Saldivar and Massoni 2018), and the organisation of units within song themes in humpback whales (Allen et al. 2019). Albert's lyrebird sequences appear to be comparable to the hierarchically organised "songs" (units/syllables within themes within songs) in humpback whales, and indeed we have made several references to the social transmission of humpback whale songs (Lilley et al. 2012, Garland et al. 2013). However, the studies on birds suggested by R2 did not investigate the function nor acquisition of higher-order sequence structure (as opposed to the order of syllables or phrases within songs), further emphasising the originality of our study.

Below we address specific comments from the reviewers.]

Reviewer(s)' Comments to Author:

Referee: 1

Comments to the Author(s)

Backhouse et al. have performed an impressive study of Albert's lyrebirds that reveals a clear temporal pattern of animal communication that is daunting to demonstrate empirically, though easily conceived on a theoretical level. I agree strongly with their insight that "while many studies treat songs as individual units of analysis, the higher-level organization of songs into sequences may too encode biologically relevant information, beyond the individual units" (lines 41-43). Indeed, for human communication it is laughable to consider the approach that communication structure could be fully understood by a narrow focus on just words (while ignoring phrases, sentences, paragraphs, etc.), yet that simplistic approach describes most of how researchers think about animal communication (with a trickle of attempts to study larger sequences, such as studies on humpback whale songs). Sequences of animal communication are probably ignored because of the empirical difficulty of the research, e.g., obtaining long sequences from several individuals, coding all song types objectively, making detailed bioacoustics comparisons. I commend Backhouse et al. for overcoming these challenges, and I look forward to applying their approaches in my own research.

[We thank the reviewer for their kind praise of this study, and hope that they find our approaches useful in their own work.]

Although I find their study impressive overall, I have a few methodological concerns, plus other general comments, followed by several specific comments:

General comment #1 – definition of 'immediate variety': For decades researchers have batted around the terms 'immediate variety' and 'eventual variety' to contrast the ways in which songbirds present their song repertoires. Backhouse et al. want to convince readers that Albert's lyrebirds present with immediate

variety, so they employ a precise definition that involves complex measurements and technical statistical analysis. The details of this definition need clarification, and to be honest, I'm not sure 'immediate variety' needs definition; the approach taken by Backhouse et al. seems more like a solution in search of a problem. Looking at Figure 1, it is obvious that these lyrebirds switch between mimetic units every few seconds, so I suspect nearly all readers will already be convinced the lyrebirds sing 'immediate variety.' I recommend dropping the empirical demonstration of immediate variety and simply referring readers to Fig 1. This would shorten the manuscript and improve readability. If the authors feel compelled to keep the technical analysis, here are a few concerns to resolve:

[Thank you for the suggestions - we address this general comment in the individual points below.]

Backhouse et al. define immediate variety based on 'repetition of units' (line 238), but it is unclear to me how they identified repetition of units. For example, mimetic unit F shown on Fig 1 always contains an element that is repeated, yet each of the five panels of Fig 1 shows just one F rather than several Fs. Why is F considered to be the cluster of elements rather than the elements themselves? Given that mimetic units can apparently include repeated elements, what would it even look like if F were repeated by a bird? (I illustrate this problem with mimetic unit F, but similar examples exist for many other mimetic units.) Answering these concerns will help readers to understand the assertion that only 29 of 7859 mimetic units were repeated (line 305).

[This is a good point. We have clarified how we identified repetition of units, and explained why we consider units with repeated elements are still considered one unit (lines 175 - 177): "Where units are composed of repeated elements and follow how the vocalisation is sung by the model species (e.g. vocalisation "F" in Figure 1), the group of elements was considered a single unit."]

I also worry there might be a flaw in the underlying assumption of lines 237-239, in which the authors calculate 'the highest chance of the repetition of units as one divided by the total number of unit types.' If I'm not misunderstanding, this calculation seems to rely on a null assumption of the random ordering of units – an assumption that is unlikely to be met in most systems of animal communication, given that most (all?) species produce some units much more frequently than other units. Once production frequencies are taken into account, the null expectations for repeating units would be lower, which would make it less likely for the authors to reach the conclusion of immediate variety.

[We agree with the reviewer's concerns, and in response have included a new analysis to take into account the different production frequencies of each unit within birds, described in lines 242 - 249. This new analysis uses the sequences prepared for the LSI analysis, and uses a randomisation procedure to test whether units are immediately repeated within sequences less than expected by chance. This new test therefore takes into account production frequencies of units within birds, and tests whether units within sequences are overdispersed.]

In sum, defining 'immediate variety' in a precise way might provide slight marginal gains, but it also provides non-trivial marginal costs because of theoretical concerns about the underlying assumptions and technical concerns over how (and whether) to apply their methods to other species.

[We appreciate the reviewer's concerns about the analysis, as it has led us to include what we believe is a more robust analysis that could easily be applied to other species. Upon reflection, we have decided that including this analysis is important because immediate and eventual variety are well established terms in birdsong research that have rarely been defined quantitatively. Our new method shows that not only do lyrebirds very rarely repeat units (as seen in Figure 1), but that units are in fact overdispersed within sequences (ie. their variability is greater than expected from chance), further strengthening the conclusion of our study. We suggest that future comparative studies on song organisation would benefit from using quantitative methods of defining immediate variety and our new analytical tool could assist in this regard (see also R1's comment about comparing mimetic song among avian vocal mimics).]

I was also surprised when the authors asserted that another prominent vocal mimic, the northern mockingbird, uses eventual rather than immediate variety (lines 403-405) – an assertion not shared by naturalists and researchers who are familiar with mockingbird song. Based on what I have read and observed about mockingbirds, I would argue their songs use immediate variety, similar to what is shown in Fig 1 for Albert's lyrebird.

[We have removed the assertion of mockingbirds singing with eventual variety and apologise for this misunderstanding. It appears the singing style of northern mockingbirds may be classified as either eventual or immediate variety depending on the level of analysis. Mockingbirds appear to repeat "syllables" (Roeske et al. 2021) or "song types" (Derrickson and Breitwisch 1992, Gammon and Altizer 2011) several times before switching; however, if each set of repeated syllables is considered a song ("phrase": Roeske et al. 2021) then it appears mockingbirds sing with immediate variety. This difference in singing style depending on unit classification highlights both the value of quantitative definitions of immediate and eventual variety, and the general importance of using consistent terms across studies (as emphasised in Odom et al. 2021).]

General concern #2 – LSI methods: Given our primitive understanding of sequences in animal communication, the study of Backhouse et al. is likely to be replicated in other species, so it is important to explain their methods clearly. Based on the work of Kershenbaum and others, Backhouse et al. recommend the Levenshtein Similarity Index (LSI) as the best way to quantify sequence similarity. I'm ok with that judgment, but I had a difficult time seeing the connections between their descriptions of how to make LSI measurements (see lines 193-204) and Figure 1. For example, when I compare Panel a and Panel e of Fig 1, I calculate a Levenshtein distance of 4 (delete V, add A, delete g, delete G), and an LSI of $1 - 4/\max(15,13) = 1-4/15 = 0.733$, not 0.765. I suspect I am missing something, but I recommend the authors illustrate their LSI calculations using an example from Fig 1.

[Well-spotted and our apologies, that was a mistake on our behalf. That number is an LSI between the sequence in Panel e and another sequence from that population. The figure has been amended to show the correct LSI. We have also added the phrase on line 207: “(example LSIs shown in Figure 1)”.]

General concern #3 – more natural history needed, especially early in Methods: I’m always a big fan of including more natural history in these types of studies because of the critical role natural history plays in constraining study design, methods and interpretation of results. At the same time, I know journals like Proc B are going to want a streamlined study, so I will prioritize my list of what natural history is lacking based on the relative importance of items within this list.

[Thank you for the suggestion - we have included more natural history as outlined below.]

Most importantly, to what extent are ‘extended singing bouts’ (line 140) a multimodal display? The remainder of your study simplifies these displays down to a sequence of mimetic units, so I want to understand what information might have been left out of your analysis. Is it possible that dance components or other visual signals also form stereotyped sequences that might be integrated with the stereotyped sequences of mimetic units?

[Albert’s lyrebirds often incorporate a visual element when singing the recital mimicry, although this part of the display is less dynamic than that used during the gronking song, and does not appear to form a specific sequence. We have included a sentence in the methods (lines 145 - 147) about the dance movement commonly accompanying the recital mimicry: “During the recital mimicry, lyrebirds often hold their ornate tail over their head and sway side-to-side, but there are no indications that recital mimicry and visual elements are coordinated.”]

When/how could social learning take place within Albert’s lyrebirds? Social learning of vocal sequences requires listening and social reinforcement, but apparently adult males live an average of 1.7km apart from other adult males (line 228) – much too far for neighbors to listen to each other. I’m sure there are contexts in which social learning could take place, but these contexts are not currently described in the paper. It would therefore be great to add a few sentences to the Discussion, e.g., right after line 357.

[We apologise for the confusion with distances. The distances listed in the methods are average distances between study individuals, which are not all neighbours. The average distance between directly neighbouring birds measured while collecting data for this study is ~300m, which is close enough for at least some song to be heard. We have specified the distance between males in lines 121 - 123: “...with approximately 300 m between neighbouring individuals (FB unpublished data).”]

More background is needed on the noise experienced by singing Albert’s lyrebirds (and therefore encountered by the field researchers). Noise can severely disrupt any study that measures acoustic similarity and is often prominent when researchers record birds from a great distance (15-30m from focal

bird – see lines 152-154). There was apparently enough noise to force you to discard an unspecified number of sequences from your acoustic analysis (line 243), and it also prevented you from measuring aggregate entropy and power (lines 249-250). Any evidence that distortions from noise might have altered your results?

[We have cleared up a few things here. It may have been misleading to say we removed sequences, as we had not specified sequences when taking acoustic measurements. Instead, we ignored recordings, or parts of recordings, that had higher levels of background interference (now specified in lines 253 - 254: “Lower quality recording excerpts were discarded...”). We have also clarified that since measurements are only compared within sequences, which are usually less than a minute long, remaining differences from noise interference are likely to be negligible (lines 260 - 263: “While aggregate entropy and power are sensitive to recording conditions, these measurements are valid for comparisons within sequences because any changes within the small timeframe of a single sequence due to recording conditions are likely to be small and random across units”).]

In addition, lyrebirds are very loud vocalists, and a distance of 15m leads to a very high SNR from the lyrebird. Since all parts of recordings with noise interference from wind, rain, traffic, or other birds were removed for this analysis, we expect that all acoustic measurements taken are valid for comparison within sequences. This is now specified in lines 154 - 155: “Albert’s lyrebirds are loud vocalists, and so this distance is usually sufficient for high quality recordings.”]

What do we know about model selection in Albert’s lyrebird? If lyrebirds are trying to highlight immediate acoustic variety, then they might sample models with acoustically-bizarre songs, while not sampling model species with acoustically bland songs. I would share examples, but I’m unfortunately not familiar enough with Australian species. (If a rigorous study of model selection has not been performed for Albert’s lyrebird, then I think you should recommend such a study – perhaps around lines 421-424.)

[Unfortunately there is almost no information on model selection in Albert’s lyrebirds, beyond an anecdotal list of the species they do commonly mimic. Recommending a study on model choice is a nice idea, and we have modified lines 430 - 436 to include this: “Instead, Albert’s lyrebirds may exhibit their mimetic abilities by mimicking accurately a limited number of particularly acoustically arresting heterospecific vocalisations, something that requires a study on model choice to confirm”.]

Do female Albert’s lyrebirds sing? If so, do they mimic?

[Yes, females sing and mimic, but there are no formal descriptions of female song. We have observed mimicry by females of a handful of heterospecifics, although it does not appear to be cyclical like the mimicry of males, and they seem to have their own whistle song as well. In superb lyrebirds, females sing in the contexts of nest and territory defence (Dalziell and Welbergen 2016a). Female Albert’s are likely doing the same thing. We have included a brief sentence about female song in lines 124 - 125: “Females also sing and produce accurate vocal mimicry, although they do not mimic as extensively as the males.”]

In line 140 'extended' could mean a minute up to several hours. Can you be any more precise?

[Overall display bouts can last for over an hour, uninterrupted mimicry usually for several minutes. This is now included in lines 119 - 121: "...Albert's lyrebirds perform dance-like displays in conjunction with their own song and sequences of vocal mimicry of other species in performances lasting from several minutes to over an hour."]

Please clarify the percentage of acoustic units within an extended singing bout that are mimetic vs nonmimetic. Fig 1 looks like at least 90% mimetic, with the only nonmimetic signals being whistle songs and gronking. Is that correct?

[While selecting the data for this analysis we did not include parts of recordings that had a high number of whistle songs (added to lines 163 - 164: "...recital mimicry containing minimal whistle songs and gronking was selected..."), and so could not give an estimate of the percentage of units that were mimetic vs nonmimetic using these data. Instead, we scored a continuous excerpt of mimicry from each population and used this to estimate that singing bouts are made up of at least 90% mimetic units (line 137: "...and makes up at least 90% of vocalisations during a display bout"; Supplementary material table S1.)]

Do whistle songs and gronking occur as repertoires? Your analysis currently focuses on just the mimetic units, and one of your main conclusions is that sequential mimetic units have high acoustic contrast. If whistle songs and/or gronking consists of repertoires, then these signals might potentially contribute to the presentation of acoustic variety.

[It is highly unlikely that gronking contributes to the presentation of acoustic variety for two reasons. Gronking is long (>10 s) and highly stereotyped with only one apparent variant in each population, and is temporally discrete, occurring only occasionally in display bouts and often followed by a pause in singing. The whistle song is also unlikely to contribute to the presentation of acoustic variety as it only occurs every 3 minutes or so (from our observations) and each male produces 1-6 variants of the whistle song (Backhouse et al. 2021), in contrast to 15-37 mimetic units. Nonetheless, we included whistle songs in the acoustic contrast analysis as they occur multiple times within a 15 minute bout, and have now clarified that in lines 256 - 257: "...we drew selection boxes around each mimetic and non-mimetic unit (including whistle songs) on the spectrogram...".]

General concern #4 – Duration results need more attention: The data of Table 3 (and Table S7) show convincingly that temporally-adjacent mimetic units are acoustically-dissimilar with regard to most of their acoustic measurements. This is an important result, particularly because it highlights what is likely a key difference in the songs of Albert's lyrebird and the northern mockingbird (see Roeske et al. 2021). That said, I find it interesting that the duration results are of comparable magnitude, but in the opposite direction. This result is mostly ignored in the Discussion. I am wondering what might be driving this result (e.g., respiratory constraints? – see the work of Franz Goller and others). I also wonder whether the

authors might need to soften their conclusion, given that four of their seven measures are all frequency measurements (likely correlated, plus frequency measurements are often distorted by variation in noise – see lines 242-252 combined with the research of Sue Ann Zollinger), and given that many potentially useful acoustic measurements, such as spectral continuity and frequency excursion, were apparently not measured.

[We thank the reviewer for picking up on the duration results. We had been unsure whether to discuss these results. We have now added a short discussion about the duration to lines 407 - 409: "...with the exception of duration, which was similar between consecutive units. Differences in temporal measures may be more difficult for females to perceive, or may be less stimulating than differences in other spectral measures." We find it unlikely that respiratory constraints would drive similarities in duration, as there are often short (~1 s) pauses between consecutive units.]

In terms of the other acoustic measurements, while there is some correlation between raw measurements of the mimetic units, correlations in differences between units is not overly high (the highest is 68%, between peak freq and low freq - correlation between peak freq and high freq is 42%). It is unlikely these results are adversely affected by noise, given we only used high quality recordings for this analysis (see lines 154 - 155 and above comment about acoustic interference).]

General concern #5 – suggestion of future studies: Although the discussion was well-written, I felt it was a little too focused on just Albert's lyrebirds. Given the ground-breaking approach of Backhouse et al., I recommend broadened the taxonomic focus within the Discussion. For example, is it possible to connect more to studies of song/call sequences performed in humpback whales, hyraxes, marsh wrens, etc.? I also recommend broadening their disciplinary focus. For example, the methods described in lines 179-184 address a challenge like the problem faced by computational biologists when they categorize DNA sequences within a chromosome, which consists of millions of nucleotides. (If you want to go even broader, your challenge is also like that faced in business and media analytics, in which researchers look for patterns in massive streams of data.) What methodological strategies have been used successfully by computational biologists that might be applied to the study of animal communication sequences? For example, might there be a song sequence in Albert's lyrebird communication that is analogous to the 'promoter' sequences that identify the start of genes?

[We thank the reviewer for their suggestions on broadening the discussion. We have now replaced the original conclusion paragraph with a discussion on broader implications (lines 442 - 455).]

One final recommendation for a study to recommend within the discussion: I think the time is ripe for a comparative study of song structure in prominent vocal mimics, such as European starlings (Goller), icterine warblers (Riegert), northern mockingbirds (Gammon), greater racket-tailed drongos (Goodale), marsh warblers (Dowsett-Lemaire), satin bowerbirds (Coleman), superb lyrebirds (Dalziell), and perhaps other mimicking species. In the past few decades, researchers in vocal mimicry have often blazed their own trails, without giving much attention to replicating the methods used in other studies. A comparative study would help to crystalize similarities and differences between mimicking species.

[Great idea! Lines 436 - 440 now state “Furthermore, higher-order organisation of mimetic song has received little attention across mimicking species, with the exception of studies on northern mockingbirds, and a comparison of sequence organisation across mimicking species would greatly help in understanding both the drivers of model choice and the function of vocal mimetic behaviours”]

Specific concerns:

Line 38: Given the general readership of Proc B, I recommend you start the manuscript with a sentence or two about the importance of higher-order structure in human communication.

[Thanks for the suggestion - the opening sentence is now about human language and music (lines 38 - 41: “Human language and music are fundamentally composed of higher-order sequences: phonemes are organised into words, which form sentences that can be structured into narratives, and notes are organised into motives and phrases, which in turn are organised into songs and other complex musical compositions.”)]

Line 39: revise ‘avian communication’ to ‘avian acoustic communication’

[Agreed - done]

Line 62-63: I recommend rewording ‘singing with little to no immediate repetition of song or unit types.’ Otherwise some readers might misinterpret your meaning, because Fig 1 shows plenty of repeated syllable units within a single ‘mimetic unit’ (see General concern #1).

[We have removed “or unit” to make this more clear (line 68).]

Lines 131-135: Consider adding a map of the five study sites as a supplementary figure, or referring to a map in another manuscript (perhaps Putland et al. 2006?).

[Thank you for this suggestion. We have added and referred to a new map in the Supplementary material (lines 126 - 127: “We studied Albert’s lyrebirds in five different sites that encompass the species’ range (see Supplementary material, Figure 1 for map)...”.)]

Line 151: In superb lyrebirds adult males look and sing differently than sub-adult males (see Zann & Dunstan 2008). Is this also true for Albert’s lyrebirds? If so, how did you account for these developmental differences in the design of your study?

[All study individuals had adult plumage and so were assumed to be mature adults. We have added a sentence to address this comment on lines 157 - 158: "...and adult male plumage confirmed either in person or from camera trap footage from display platforms."]

Line 164: Does 'where possible' mean 19 of 25? If so, please add the numbers?

[We mean 20 out of 25, where data for the remaining 5 birds are taken from two days. This is now clarified on line 167: "For 20 out of 25 focal birds, this was continuous mimicry from one recording."]

Line 179: I recommend adding a phrase, so that it reads "...structured into bouts that can last for several minutes." ('Bout' is defined differently by researchers studying other songbirds, so the added phrase clarifies how you define 'bout'.)

[Here we have specified that sequences are further structured into bouts "lasting from several minutes up to over an hour" (line 182).]

Line 183: I'm splitting hairs here, but the satin bowerbird vocalization on the left side of Fig 1 doesn't look to me like a 'long' buzz. Rather, it looks like a short buzz that gets repeated many times. (It's probably fine with me if you ignore this comment.)

[Good point that this vocalisation is not one long buzz. This was simply a name used to remember unit types. As the name of the vocalisation is not important for this analysis, we have rephrased to refer to "satin bowerbird vocalisation 'd' in Figure 1..." (lines 186 - 187) to avoid confusion over naming schemes.]

Line 191-192: Do the parenthetical numbers refer to mean +/- one standard deviation?

[That is correct. This has now been clarified (line 195).]

Line 216: If I understand correctly, an AMOVA functions like a nested ANOVA. If this is true, I recommend you add a short phrase here to clarify, because many (most?) birdsong readers will be unfamiliar with AMOVA. I initially thought it was a misprint.

[Yes, that is right. We have now clarified this on lines 220 - 221: "AMOVA functions like a nested ANOVA and was developed to measure the amount of genetic variation..."]

Line 260-261: Forgive me for my statistical ignorance, but is 'standard deviation of the mean' the same as 'standard error?'

[Apologies for the confusing wording. We have removed "of the mean" (line 273).]

Line 268: I think you meant 'an,' not 'on'

[We have rephrased this line to clarify our meaning: "...and contained mimicry of 4.80 ± 1.72 heterospecifics on average..." (line 280).]

Line 286: More context needed for the three Greek term listed in Table 2. I assume these three numbers somehow connect to the information presented in Table 2, but I don't yet see the connection.

[These numbers are the test statistics (analogous to F-statistics). They have been moved into the table, but definitions retained in the table caption, for clarity (table on line 299).]

Line 288-292: What comparison(s) do you want readers to make for Figure 2? If I understand correctly, the main story of Figure 2 is that each of the three LSI averages is considerably higher than the three expected LSI averages. If my thinking is correct, then I recommend adding three expectation lines and inserting a 'take home message' into the figure heading. Right now, the figure looks like you might be trying to highlight that the 1st and 2nd column means are significantly higher than the 3rd column mean; I'm pretty sure that's not the point you want readers to make.

[Thank you for the suggestions. Fig 2 actually was meant to draw a comparison between the three groups; however, this figure could also be used to show that real LSIs are higher than expected. As such, we have added expected values to the figure, and specified in the caption that "LSIs are different between groups, and significantly higher than expected from random" (line 304).]

Line 323: Would the 'duet codes' studied by David Logue and others be another example of social transmission of vocal sequences in birds? (In brief, Logue found that the antiphonal 'FMFMFM' duets sung by Male and Female wrens normally use codes, e.g., female song #12 always alternates with male song #26 and no other male song type, female song #4 alternates with male song # 18 and no other male song type.) I'm guessing you would not consider these as vocal sequences and that you could leave the manuscript intact, but I thought I'd raise the question, just in case.

[This is really interesting work, but it is a bit different - it looks like the duet codes are very specific to each pair, or even just individual females. So there is some level of social coordination, but there is not a set "code" that is shared within a population like there is in Albert's lyrebirds. We have decided to leave duet codes out of the manuscript to avoid overcomplicating the discussion.]

Line 345-357: Great reasoning; this paragraph felt triumphant.

[Thank you for this kind praise.]

Lines 385-390: These two rules seem well-supported by your data, but I don't see how the two rules by themselves would lead to the similarity of sequences across all the geographic range.

[If lyrebirds are mimicking similar species across the range, these rules mean that where the same vocalisations are mimicked they are likely to be presented in a similar fashion. However, we realise that species-wide similarities may simply be a result of social transmission. In addition, the opening sentence to this paragraph is not particularly relevant to the following section, and so we have rephrased to focus just on the presence of those two rules (lines 399 - 400).]

Lines 397-398: What exactly does it mean to 'abridge mimetic units by removing internal repetition within imitated sounds?' Does this mean that if a superb lyrebird wants to imitate a multi-syllable model vocalization, that it only one of the syllables within the model vocalization while dropping the other syllables?

[Where there are repeated elements within the original vocalisations, the repeats have been dropped, so that there is only one rendition of each element type within the imitated vocalisation. We have rephrased to clarify this in lines 413 - 416: "Further, superb lyrebirds appear to abridge mimetic units by removing repetitions of certain elements within imitated multi-element sounds, while maintaining the original element order and number of element types..."]

Line 439: I think you mean 'Special' not 'Species'

[Well-spotted, thank you!]

Referee: 2

Comments to the Author(s)

This manuscript looks at the structure of lyrebird songs, a species that mimics other species and includes some of these mimetic songs into their own song. The authors find that lyrebirds produce stereotyped song sequences, where the also the mimetic elements occur in stereotyped sequences. Given that these sequences are similar within populations, the authors argue that this reflects social transmission. Moreover, these sequences maximize the contrast between the elements.

The use of a species that mimics other species provides a nice opportunity to study song structure. Clearly, this is trickier in species that do not mimic other species, but it is still possible. I think that there have been more studies on song that did look into the song sequences, and how this is socially acquired, at least what I could see when scanning for literature. Sure, this must be studied in zebra finches too?

Although lyrebird song includes mimetic elements, the structure of the song seems to be acquired socially. I therefore wonder how these mimetic elements are learned ontogenetically. Are they learned from other lyrebirds, or picked up from the species that are mimicked? Given that the species which are mimicked do not occur everywhere in the study population, I speculate that they are also socially transmitted, along with the structure of the song. Thus, although these elements are originally copied from other species, if they are learned mostly from conspecifics, then it's strictly speaking not mimicry anymore.

I assume the authors do not have any genetic and/or life-history information about the individuals used in the study? These data would be helpful to support the argument of the authors.

Please see in the enclosed PDF for more comments.

[We thank the reviewer for their helpful suggestions on the manuscript, which we have used to improve the study. We disagree with a couple of points, however, which may be due to poor explanations on our part. We have attempted to clarify these points here and in the manuscript.]

The first point is whether social transmission of sequences has already been shown in birds, possibly zebra finches, and we thank the reviewer for their suggested references listed below. It is true that social transmission of sequences of syllables within songs has been demonstrated in other species (e.g. Menyhart et al. 2015), and that overall ordering of vocal units, whether it is syllables within songs (Cody et al. 2016, Zsebők et al. 2020, Paul et al. 2021), or song within song bouts (Benitez Saldivar and Massoni 2018), has received some attention in the literature. The references listed by R2 below support a general interest in sequences in bird song. However, we are focussed on the social transmission and function of sequences of multi-syllabic songs within song bouts, which we believe is novel because it pertains to the level of song organisation above those addressed previously. To resolve this issue it is important to use the correct terminology, and we may have been unclear with our terminology in the manuscript. We attempt to follow the terminology of Odom et al (2021) with elements and syllables organised into songs, and songs into song bouts. We have seen "song sequence" used to describe sequences of syllables within songs in the literature (Troyer and Doupe 2000, Fishbein et al. 2020), but in Odom et al (2021), song sequence is synonymous with song bout, and it is this definition of song sequence, meaning sequences of songs, that we are focussing on. To clarify this, we have added the term "higher-order" to our title, equated "song sequence" with song bouts in lines 45-46, and increased our use of the term "higher-order" to distinguish the sequences of mimicry investigated in the manuscript from sequences of smaller vocal units in other studies (e.g. lines 28, 53, 338, 436, 443, 447, 454).

The second point is whether mimicry of Albert's lyrebirds can in fact be classified as mimicry if it is learnt from other lyrebirds. This is a very interesting discussion point; however, we believe that the heterospecific vocalisations sung by Albert's lyrebirds should be considered mimicry for two reasons:

We use a widely employed definition of mimicry that is not dependent on the mode of acquisition, nor the sensory modality through which it is expressed (e.g., see Dalziell and Welbergen 2016b, Jamie 2017,

de Jager and Anderson 2019). Classical examples of static visual mimicry (e.g., Batesian mimicry of butterfly wing patterns: Nishikawa et al. 2015) may be acquired from conspecifics (through genetic inheritance), and under the current theoretical framework, vocal mimicry is no different (Wickler 2013, Dalziell et al. 2015). Therefore, concluding that a signal is not mimetic because it is inherited from conspecifics, as suggested by R2, would actually depart from conventional mimicry theory.

Lyrebirds almost certainly learn their mimetic sounds from both the models and other lyrebirds (Putland et al. 2006), something that may occur in multiple mimicking species (e.g. starlings: Hindmarsh 1984, Galerida larks: Laiolo et al. 2011). In addition, for the bowerbird vocalisation to be included in the repertoire of lyrebirds, it must have been learnt from bowerbirds to begin with, as the mimicked vocalisations are too different to lyrebird-specific vocalisations to have acoustic similarities by chance.

To clarify our use of the term mimicry, in line 77 we refer to Dalziell et al (2015), where “a vocalisation is mimetic if the behaviour of the receiver changes after perceiving the acoustic resemblance between the mimic and the model, and this behavioural change confers a selective advantage on the mimic”. We have also stated more clearly that mimicry is likely learnt from both conspecifics and heterospecifics in Albert’s lyrebirds (line 96: “...indicating some level of learning from heterospecifics”; lines 139 -141: “Mimicked vocalisations appear to be learnt from both conspecific tutors and heterospecifics models, though it is unclear how these two mechanisms interact.”; lines 379: “...and learn mimetic units at least in part from heterospecific models.”

Further comments from the reviewer are addressed below.]

Line 16: this is not considering the recent advances in communication based on calls

[It is true that there have been some recent advances in syntax/sequences in calls. However, calls tend to be comparatively short and simple, with studies focussing on transitions between elements or syllables (e.g. Freeberg and Lucas 2012, Suzuki et al. 2018), in contrast to the extended sequences of multi-syllabic units sung by Albert’s lyrebirds. We have therefore left this sentence in its original form.]

Line 28: has this really not been studied in model organisms eg Zebra finches?

[While zebra finches have been shown to socially transmit the order of syllables within songs, here we study the organisation of whole songs into song sequences. We have now specified that our subject is “higher-order sequences of songs” (line 28).]

Line 35: what does complexity mean here?

[We have now added a phrase to line 64 to state our definition of complexity as: “the number and acoustic diversity of syllable or song types in and individual’s repertoire”. This takes into account an individual’s ability to sing (or mimic) a large number of acoustically diverse songs.]

Line 45: there are more species where this has been studied, it appears:

“Structure, syntax and small-world organisation in the complex songs of California Thrashers (*Toxostoma redivivum*)”

“Network analysis reveals underlying syntactic features in a vocally learnt mammalian display, humpback whale song”

“Song structure and syllable and song repertoires of the Saffron Finch (*Sicalis flaveola pelzelni*) breeding in Argentinean pampas”

“Characterising the flight song: repeatable individual variation of Ovenbird song features”

“Sequential organisation of birdsong: relationships with individual quality and fitness”

[We thank the reviewer for these interesting references; however, they do not address the causes of higher-order organisation of songs. We have rephrased the addressed line in the text to clarify our meaning (lines 47 - 48: “However, the proximate and ultimate causes underlying the organisation of sequences of songs remain unclear in all but a handful of species). Please refer to the general to Reviewer 2 on the social transmission of higher-order sequences for more detail.]

Line 52: Cultural and genetic evolution in mountain white-crowned sparrows: song dialects are associated with population structure

[This is a great paper, but shows social transmission of songs, not song sequences. We have reinforced some terms in this paragraph to confirm that we refer to sequences of songs, not sequences of elements/syllables within songs (line 53: “higher-order song sequences”; line 59: “individual songs”).]

Line 60: How can we ensure that we can judge this for birds? I think one would need playbacks to proof this

[We agree, we do not know for certain that enhancing perceptions of repertoire complexity would have an impact on reproductive success without direct experimental evidence. Nevertheless, since in some species males with larger or more varied repertoires have been shown to have a higher reproductive success, it is highly likely that females are judging their vocal abilities. It then follows that males should organise their repertoire in a way that emphasises the traits females prefer. We have added the word “likely” to line 66 to soften our statement (“organisation song sequences in a way that enhances receiver perception of complexity would likely benefit reproductive success”).]

Line 66: please use a more specific word, it is unclear in which respect “complex” is used

[Please see above comment - we have now defined complexity as “the number and acoustic diversity of syllable or song types in and individual’s repertoire” (line 64).]

Line 77: I am not fully convinced here. What about the units in there? Also, in general I think there are methods that can do that reliably, using statistical inference on what a unit is

[Thanks for this point. It is true that units in conspecific song can be clearly defined for some species, and units in some mimicry can be unclear. In response, we have simplified the paragraph by removing this argument (lines 76 - 77).]

Line 86: if this MS here is not tackling these issues, I would mention them only in the discussion

[Thank you - this has been removed (line 83).]

Line 93: meaning? It's a vague word

[Complexity - "the number and acoustic diversity of syllable or song types in and individual's repertoire" (line 64)]

Line 95: sure, but the overarching question is of that is happening for bird song, and the answer would be yes.

[Thank you for drawing attention to this. We have clarified throughout the text that we are referring to "higher-order sequences". Please see our general response to Reviewer 2 above regarding the novelty of social transmission of higher-order sequences.]

Line 100: again. It is established for song that this is happening.

[Please see above general response outlining the novelty of showing social transmission of higher-order sequences.]

Line 104: if these mimicry elements are spread socially, then it's not mimicry anymore. It's just learning the song elements of another individual

[Please see the general response to Reviewer 2 regarding mimicry. By the established definition that we are using, this should still be considered functional mimicry.]

Line 105: what is the mechanism of transmission: parent (father) - offspring, or displaying male - subordinate males or other adult displaying males?

[We can assume that juveniles or subadult males are learning from both related and unrelated adult males, as with female-only parental care, lyrebirds are unlikely to know who their father is. Unfortunately we do not know the age of learning in this species. This is a fascinating question that we are currently trying to address experimentally, but it will take years to get an answer. We have included a sentence about song learning on lines 123 - 124: "Males have no role in parental care, and so display components may be learnt from both related and unrelated individuals".]

Line 151: do you know the relatedness among these males and previous life history?

[Unfortunately, our study system is not logistically readily amenable to gathering this type of information, but information on the relatedness and the history of individuals is not essential for our conclusions - only their display locations and that they are adult males (lines 157 - 158: "...and adult male plumage confirmed either in person or from camera trap footage from display platforms".)]

Line 265: see above, if they are learned from other conspecifics then it's not mimicry anymore.

[Please see our general comment above regarding the definition of mimicry]

Line 294: how can this result be in line with the idea that the sequences is transmitted socially?

[While somewhat confusing, this does not necessarily refute the hypothesis of social transmission of sequences, as we have explained in the discussion. Our inclusion of a hypothesis on changes with geographic distance in the introduction may have made things confusing. We have now removed that hypothesis, but retained this result as it is an interesting result, both since other lyrebird vocalisations change with geographic distance (whistle songs - lines 387 - 389), and as the geographic variation of mimicry or song sequences has rarely been described in other species. We have further explained that this result may be due to subtle, unaccounted for differences in the abundance of model species at each site (lines 365 - 366: "...albeit we could not account for possible differences in the local abundance of model species."; lines 392 - 393: "...the recital mimicry reported in the present study could be affected by subtle differences in the local availability of model species...").]

Line 300: meaning? It is unclear whether this means that all individuals have the change to learn it directly, or that the "mimetic" parts of the song are also learned socially from conspecifics?

[We apologise that this was not clear. We have now added a sentence to the methods saying that we considered a species likely to be in the area if the value of the SDM at that point is 0.5 or greater (lines

239 - 240: “Species were deemed likely to occur at a site if the value of the SDM at that site was 0.5 or greater”). Therefore all individuals should be able to learn directly from most (if not all) models.]

Line 322: please see above

[We are not sure what this comment refers to. By our definition of mimicry, these are sequences of mimicry, and to our knowledge whole sequences of mimicry have not been quantified in any study previously.]

Line 333: I would remove them. Alarm calls are a different story.

[Agreed. We have removed this, and moved the line about whale songs to fit the paragraph better (lines 345 - 346: “Where similar methods have been used, humpback whales have greater song similarities than Albert’s lyrebirds...”).]

Line 359: That’s a question of definition. The old work on song sparrows shows this too.

[To our knowledge the work on song sparrows concerns the organisation of syllables within songs, not extended sequences of whole songs. We have clarified that we are talking about sequences of songs (lines 370 - 371: “To our knowledge the sharing of whole, stable sequences of songs between local individuals has rarely been shown in wild populations of birds or mammals”).]

Line 373: I guess this is simply a consequence of the rather patchy habitat and the limited dispersal options.

[We agree that this is possible, as geographic patterns in vocal behaviours can be complex. We also realise it may be due to small differences in model species assemblage, as there is also no signal of geographic distance in a mimetic component of another vocalisation sung by Albert’s lyrebirds. We now explain this in lines 390 – 394 (see above comment about geographic variation).]

Line 393: complex in which respect? Also, could this not also just reflect a mechanic constraint of song production?

[We have now clarified our definition of repertoire complexity as “the number and acoustic diversity of syllable or song types in an individual’s repertoire” (line 64), as lyrebirds may be showing both the repertoire size and the variation in the sounds they can produce. We are not sure how differences in acoustic measures would reflect mechanistic constraints. This is more likely to have an impact on very rapid changes in frequency (such as in trills), not between vocalisations of up to a few seconds long, with silences of ~1s between them (Figure 1).]

Line 411: this is a review paper, are the specific papers that show this?

[We apologise for this confusing sentence. The review paper was cited there because those are traits that may be sexually selected for in mimicking species more generally. We have now removed that part of the sentence (lines 428-429), and focussed on what may be selected for in lyrebirds given the evidence from this and other studies.]

Line 413: that could be argued for song in general. Males (or females) that sing not accurate enough may not be perceived as high quality individual.

[Good point! However, we prefer to keep this paragraph focussed on mimicry in order to streamline the discussion.]

Line 420: all interesting, but I think one would need experiments to underpin these arguments.

[Yes, more, particularly experimental work would be needed for sure. We have included suggestions for future work on lines 435 - 440: "...something that requires a study on model choice to confirm. Furthermore, higher-order organisation of mimetic song has received little attention across mimicking species, with the exception of studies on northern mockingbirds, and a comparison of sequence organisation across mimicking species would greatly help in understanding both the drivers of model choice and the function of vocal mimetic behaviours"]

Literature cited in response

- Allen, J. A., E. C. Garland, R. A. Dunlop, and M. J. Noad. 2019. Network analysis reveals underlying syntactic features in a vocally learnt mammalian display, humpback whale song. *Proceedings of the Royal Society B* **286**:20192014.
- Backhouse, F., A. H. Dalziell, R. D. Magrath, A. N. Rice, T. L. Crisologo, and J. A. Welbergen. 2021. Differential geographic patterns in song components of male Albert's lyrebirds. *Ecology and evolution* **11**:2701-2716.
- Benitez Saldivar, M. J., and V. Massoni. 2018. Song structure and syllable and song repertoires of the Saffron Finch (*Sicalis flaveola pelzelni*) breeding in Argentinean pampas. *Bioacoustics* **27**:327-340.
- Cody, M. L., E. Stabler, H. M. Sanchez Castellanos, and C. E. Taylor. 2016. Structure, syntax and "small-world" organization in the complex songs of California Thrashers (*Toxostoma redivivum*). *Bioacoustics* **25**:41-54.
- Dalziell, A. H., and J. A. Welbergen. 2016a. Elaborate Mimetic Vocal Displays by Female Superb Lyrebirds. *Frontiers in Ecology and Evolution* **4**.
- Dalziell, A. H., and J. A. Welbergen. 2016b. Mimicry for all modalities. *Ecology Letters* **19**:609-619.
- Dalziell, A. H., J. A. Welbergen, B. Igic, and R. D. Magrath. 2015. Avian vocal mimicry: A unified conceptual framework. *Biological Reviews* **90**:643-668.
- de Jager, M. L., and B. Anderson. 2019. When is resemblance mimicry? *Functional Ecology* **33**:1586-1596.
- Derrickson, K. C., and R. Breitwisch. 1992. Northern mockingbird. *The Birds of North America, Inc.*
- Fishbein, A. R., W. J. Idsardi, G. F. Ball, and R. J. Dooling. 2020. Sound sequences in birdsong: how much do birds really care? *Philosophical Transactions of the Royal Society B* **375**:20190044.

- Freeberg, T. M., and J. R. Lucas. 2012. Information theoretical approaches to chick-a-dee calls of Carolina chickadees (*Poecile carolinensis*). *Journal of Comparative Psychology* **126**:68.
- Gammon, D. E., and C. E. Altizer. 2011. Northern mockingbirds produce syntactical patterns of vocal mimicry that reflect taxonomy of imitated species. *Journal of Field Ornithology* **82**:158-164.
- Garland, E. C., M. J. Noad, A. W. Goldizen, M. S. Lilley, M. L. Rekdahl, C. Garrigue, R. Constantine, N. Daeschler Hauser, M. Michael Poole, and J. Robbins. 2013. Quantifying humpback whale song sequences to understand the dynamics of song exchange at the ocean basin scale. *The Journal of the Acoustical Society of America* **133**:560-569.
- Hindmarsh, A. M. 1984. Vocal mimicry in starlings. *Behaviour* **90**:302-324.
- Jamie, G. A. 2017. Signals, cues and the nature of mimicry. *Proceedings of the Royal Society B: Biological Sciences* **284**:20162080.
- Laiolo, P., J. R. Obeso, and Y. Roggia. 2011. Mimicry as a novel pathway linking biodiversity functions and individual behavioural performances. *Proceedings of the Royal Society B: Biological Sciences* **278**:1072-1081.
- Lilley, M. S., E. C. Garland, M. L. Rekdahl, M. J. Noad, A. W. Goldizen, and C. Garrigue. 2012. Improved versions of the Levenshtein distance method for comparing sequence information in animals' vocalisations: tests using humpback whale song. *Behaviour* **149**:1413-1441.
- Menyhart, O., O. Kolodny, M. H. Goldstein, T. J. DeVoogd, and S. Edelman. 2015. Juvenile zebra finches learn the underlying structural regularities of their fathers' song. *Frontiers in Psychology* **6**:571.
- Nishikawa, H., T. Iijima, R. Kajitani, J. Yamaguchi, T. Ando, Y. Suzuki, S. Sugano, A. Fujiyama, S. Kosugi, and H. Hirakawa. 2015. A genetic mechanism for female-limited Batesian mimicry in *Papilio* butterfly. *Nature genetics* **47**:405-409.
- Odom, K. J., M. Araya-Salas, J. L. Morano, R. A. Ligon, G. M. Leighton, C. C. Taff, A. H. Dalziell, A. C. Billings, R. R. Germain, and M. Pardo. 2021. Comparative bioacoustics: a roadmap for quantifying and comparing animal sounds across diverse taxa. *Biological Reviews*.
- Paul, N., M. J. Thompson, and J. R. Foote. 2021. Characterising the flight song: repeatable individual variation of Ovenbird song features. *Bioacoustics* **30**:232-251.
- Putland, D. A., J. A. Nicholls, M. J. Noad, and A. W. Goldizen. 2006. Imitating the neighbours: Vocal dialect matching in a mimic-model system. *Biology Letters* **2**:367-370.
- Roeske, T. C., D. Rothenberg, and D. E. Gammon. 2021. Mockingbird Morphing Music: Structured Transitions in a Complex Bird Song. *Frontiers in Psychology* **12**.
- Suzuki, T. N., D. Wheatcroft, and M. Griesser. 2018. Call combinations in birds and the evolution of compositional syntax. *PLoS biology* **16**:e2006532.
- Troyer, T. W., and A. J. Doupe. 2000. An associational model of birdsong sensorimotor learning II. Temporal hierarchies and the learning of song sequence. *Journal of neurophysiology* **84**:1224-1239.
- Wickler, W. 2013. Understanding mimicry—with special reference to vocal mimicry. *Ethology* **119**:259-269.
- Zsebők, S., G. Herczeg, M. Laczi, G. Nagy, É. Vaskuti, R. Hargitai, G. Hegyi, M. Herényi, G. Markó, B. Rosivall, E. Szász, E. Szöllősi, J. Török, and L. Z. Garamszegi. 2020. Sequential organization of birdsong: relationships with individual quality and fitness. *Behavioral Ecology* **32**:82-93.

Appendix B

Revision comments - Fiona Backhouse, Anastasia H. Dalziell, Robert D. Magrath, Justin A. Welbergen

Manuscript ID RSPB-2021-2498 – Higher-order sequences of vocal mimicry performed by male Albert's lyrebirds are socially transmitted and enhance acoustic contrast

Please note that all comments from the reviewers are in normal font, and our replies to the comments are in **boldface**. All changes to the text are marked as tracked changes in "ALBsequences_MAINTTEXT_R2_trackedchanges". All 'new' line numbers refer to this file with tracked changes. The clean, revised document is named "ALBsequences_MAINTTEXT_R2_clean".

Associate Editor Board Member

Comments to Author:

We have now received two reviews of your revised manuscript. Both the reviewers and I find the manuscript much improved. However, the reviewers point to areas for further improvement. These largely involve softening or clarifying some of the conclusions and further acknowledging alternative explanations and interpretations. Most notably, Reviewer 1 points out that the evidence for social transmission of sequences is weak given that similarity above chance might be the product of shared repertoires and transition tendencies (a shared tendency to increase contrast between units).

Reviewer 1 suggests testing this directly but, short of that, this weakness should be acknowledged.

Reviewer 2 has concerns about the immediate variety analysis particularly as it relates to repeated elements (like vocalization F in figure 1). Perhaps it would help to indicate that the duration and timing of repetitions in such units also support treating them as a single element (in addition to the mimicry-based criteria given in lines 175-177).

Reviewer 2 also has concerns about social transmission and the definition of mimicry. I share this concern. While I understand the utility of a broad definition of mimicry that does not imply a mechanism (as in Batesian mimics), with vocal mimicry there is an implication that there is direct copying between species (as stated in the definition of vocal mimicry on line 76-77). This definition really does focus on the mechanism of transmission (rather than the function). This seems to be a larger issue with the use of the term 'mimicry' in this context which is beyond the scope of this manuscript. However, the ambiguity with the meaning of vocal mimicry in relation to the results seems worth mentioning in the Discussion. Both reviewers make additional suggestions for clarification and improvement.

[Thank you for your time in providing this second review. We agree that the manuscript was much improved after the first revision, and feel that this second revision has improved it further, especially in clarity. We address two points specifically here, and address all further points to the reviewers' comments.

First, thank you for suggesting the additional clarification on how we treat repeated elements. We have now rephrased lines 181 - 185: "Most mimetic units contained multiple elements, and some units contained repetitions of short elements (e.g. vocalisation "F" in Figure 1). These sequences of repeated elements were considered a single unit as they comprise short phrases and mimic how the vocalisation is sung by the model species.

Second, both Reviewer 2 here and Reviewer 2 from the first revision brought up interesting discussion points about social acquired mimicry and the definition of vocal mimicry, and we thank them for

drawing attention to this issue. We agree that this is beyond the scope of the current manuscript and deserves greater attention elsewhere. We have now addressed our stance on vocal mimicry in the manuscript, which we describe in further detail in our response to Reviewer 2. We have also revised lines 76-77 as we realised this phrasing did not capture how we view vocal mimicry (lines 76-77: “One vocal behaviour in which sequences are especially overlooked is vocal mimicry”). We refer to our new clarification on vocal mimicry in our response to Reviewer 2.]

Reviewer(s)' Comments to Author:

Referee: 3

Comments to the Author(s).

This is a well-designed and well executed study, which has done a nice job of analysing song sequences in a rigorous and effective way. I'm joining as a reviewer after the first revision, so I'm afraid that I may be suggesting some changes that would have been more appropriate on first submission, but I can reassure the authors that I believe the study is well worth publishing, and hopefully the additional analysis should not be particularly onerous. Also, I am waiving anonymity, as I am recommending that the authors consult a couple of my own papers.

I am very pleased to see the extensive use of exact tests to measure the significance of the findings. If only more studies would make use of this! As not all researchers are familiar with randomisation tests, it is always worth restating the hypothesis (e.g. on line 249: that repeated units occur less likely than would be expected by chance). Table 1 could do with giving the p values from the Z test.

[We thank the reviewer for their praise of this approach. We have now included the p values in Table 1 and stated the hypotheses of the permutation tests on lines 258 - 259 (“...by testing if repeated units occur less than expected by chance.”) and lines 283 - 284 (“To test whether acoustic contrast was higher between consecutive units than expected by chance...”).]

One little quibble that I have is the use of “higher order” as a description for the patterns being studied. While I think that the term is technically accurate (first order relationships being higher order than simple repertoire diversity), it might be confused with the kinds of hierarchical structures seen in, for instance, humpback whale song. Although the authors do a good job of looking at transition probabilities, I myself would not call this “higher order”. I'm not overly fussed about this, but I think it's worth making the point that the authors are not, for instance, looking at repetition of motifs.

[We apologise that we have not been clear with our use of the term “higher-order”. We are in fact looking at similar structures to those used in humpback whale song - in Albert's lyrebirds, elements are organised into mimetic units (as copied from the model species), which in turn are organised into sequences, which are then organised into song bouts. Our unit of analysis here is the multi-element mimetic unit, as opposed to elements within units. We therefore use the term higher-order to differentiate our focus on sequences of multi-element units from other studies that investigate the organisation of elements or syllables within songs.]

We have now clarified that the mimetic units sung by Albert's lyrebirds often contain multiple elements, and so by looking at sequences of multi-element units we are considering a higher level of organisation than the order of elements within mimetic units.

Changes are on line 89 (“Males mimic vocally a number of multi-element avian sounds...”), lines 135 - 136 (“...a series of multi-element mimetic songs (or ‘units’) including mimicry of...”), and line 181 (“Most mimetic units contained multiple elements...”)]

In addition, when discussing the repetitions (line 319), it's worth mentioning the implicit assumption that the sequences are simple Markov chains, i.e. that the probability of a unit occurring can be predicted by the preceding unit(s). If there are indeed higher order structures, then this assumption is unlikely to be met. However, the observation that the repetition rate is lower than expected is an important one, as most animal vocal sequences appear to have greater repetition (see Kershenbaum et al. (2014). Animal vocal sequences: not the Markov chains we thought they were. *Proceedings of the Royal Society B: Biological Sciences*).

[We thank the reviewer for making this suggestion, particularly as many readers may be used to thinking of bird song sequences as simple Markov chains. We have now stated that we make this assumption in the methods as clarification before describing the analyses on the sequences (lines 191 - 194: “Here we assume that sequences of mimetic units are Markov chains, where the probability of each unit occurring is determined by a finite number of previous units, although there may be further complex, undetected patterns in sequence organisation.”)]

For the geographic analysis, I would refer the authors to my paper: Kershenbaum et al. (2012). Syntactic structure and geographical dialects in the songs of male rock hyraxes. *Proceedings of the Royal Society B: Biological Sciences*. This would be a relevant comparison, because the songs are group territorial songs, and therefore perhaps more likely to be perceived consistently by neighbours.

[We thank the reviewer for pointing out this valuable reference. We agree that it is possible the sedentary nature of lyrebirds prevents cultural transfer across the landscape, leading to the lack of geographic signal in sequence differences between populations. However, if this were the case then it would be surprising that variation in species-specific whistle songs between populations does show a geographic signal. We have now added a sentence on lines 424 - 426 to suggest the explanation in Kershenbaum et al. (2012): “Additionally, the lack of correlation between sequence similarity and geographic distance between populations may reflect limited cultural transfer between populations of a highly sedentary species.”]

I'm a little worried about the acoustic similarity analysis. Were the metrics standardised (e.g. to zero mean and unit SD, or alternatively to a range 0-1)? If not, it is hard to calculate mean acoustic difference, if the metrics have different orders of magnitude.

[We apologise that this part of the methods was unclear. We did not measure one average acoustic difference across all acoustic metrics, but instead measured the average difference separately for each metric individually. We have now clarified this on lines 282 - 283: “...we calculated the average difference between each consecutive unit separately for each acoustic variable.” and lines 285 - 286: “...calculated the average acoustic distance for each acoustic variable again.”]

I think that the conclusion (line 332) that there is social transmission is a weak one. There are other explanations that should be discussed. In particular, I think that the observations could be explained by a combination of the acoustic contrast rules, together with a difference in repertoires between the different

populations. I would say that it would be a very useful (and easy) additional test to see whether there is significant difference in the repertoire distributions or not. Particularly as you state (line 430) that the repertoire is limited, it seems essential to tell the readers whether the repertoires are the same (or similar) between populations.

[We thank the reviewer for this suggestion, as we can see how our description of the repertoires and the results of the acoustic contrast analysis may lead to this conclusion. However, individual Albert's lyrebirds have moderately large repertoires (at least 15 units, but up to 37 - line 375), which could lead to an extremely large combination of possible sequence structures. There are multiple ways in which a high acoustic contrast could be achieved, and in fact, while the mean acoustic contrast across sequences is higher than expected by chance, it is still not completely maximised across all sequences. It therefore seems highly implausible that individuals would converge on the same sequence structure without any social learning. Given the current length of the manuscript we feel it is more parsimonious to address this possibility in the discussion than to add an extra analysis.]

We have acknowledged the explanation proposed by the reviewer in the discussion on lines 388 - 393: "Another possible explanation is that sequence sharing is a by-product of individuals applying shared acoustic contrast rules to a shared mimetic repertoire. However, given the large mimetic repertoire of Albert's lyrebirds, there are multiple combinations of units that could lead to a high acoustic contrast within the sequence (supplementary material, figure S3), and so population differences in both mimetic repertoires and sequence structure are more likely to reflect social learning."

We have also rephrased line 430 to clarify that repertoires are smaller than lyrebirds seem capable of, rather than being of an overall limited size (lines 464 - 466: "...Albert's lyrebirds do not utilise their full mimetic repertoire during the recital mimicry, given their ability to mimic other heterospecifics during sub-song...").]

Additionally, I think the conclusion on line 377, that the birds perceive entire sequences, is also not well supported, as it hasn't been tested explicitly. Perhaps it would be possible to make this speculative claim if hierarchical structures were found, but the similarity between overall sequences could be explained by simpler, short-term rules. I think it would be good to qualify this conclusion. It's still a very interesting result!

[We thank the reviewer for this suggestion. It is true this conclusion should be tested explicitly. We have now softened our conclusion to suggest that lyrebirds are perceiving both individual units and temporal relationships between units (lines 405 - 409: "Our results suggest that during the learning process, Albert's lyrebirds not only perceive vocalisations at the level of individual units, where mimetic units may be learnt at least in part from heterospecific models, but may also perceive the temporal relationships between units when learning from conspecifics.")]

A few minor points:

Line 164: How were the songs selected?

[Mimicry was selected based on recordings that had a high signal-to-noise ratio and with minimal breaks for species-specific vocalisations. We have rephrased lines 167 - 169: "Approximately 15 minutes

of recital mimicry was selected from the recordings of each male, with effort to choose a continuous bout of recital mimicry with a high signal-to-noise ratio and with minimal species-specific song.”]

Line 184: Delete “In order”.

[Removed - thank you.]

Line 186: It is explained that sequences begin with an instance of the most common unit. Could you provide some rationale for this, as it is not immediately obvious why this should be the criterion?

[We have now added more rationale for this in lines 194- 200: “As recital mimicry is often presented for an extended period without a break, and whistle songs are sung at inconsistent points within the recital mimicry, the start and end points of individual sequences within these bouts were not obvious. To apply the same treatment to all birds and avoid creating artificial differences between populations in sequence length, we split recorded bouts into sequences using the mimetic unit that had the most consistently high occurrence in all birds...”]

Line 191: You remove whistle songs. Does this mean that units on either side of the whistle song are treated as being adjacent? This could cause problems for the sequence analysis, as they are not genuinely adjacent. Some discussion of this would be sufficient

[Whistle songs do not appear to be a consistent part of the sequence (lines 202 - 203: “whistle songs are temporally discrete within singing bouts”), however it is true that after a whistle song the sequence may continue at a different point to what would normally come next. Including whistle songs and introductory notes in the sequences makes very little difference to the overall sequence similarities, and we have added these similarities to Table 3 in the supplementary material. We have also including a brief discussion of this on lines 205 - 207: “Removing whistle songs and introductory elements marginally increasing sequence similarities (supplementary material, table S3), but this is unlikely to affect any conclusions from results.”]

Signed review:

Arik Kershenbaum

Referee: 1

Comments to the Author(s).

I was Reviewer #1 on the original submission. After spending several hours looking through the comments of the editor, me, and R2, and the detailed responses of Backhouse et al., I can only conclude the authors did a thorough job of addressing our criticisms. I still maintain a couple of important disagreements (see below), but these come down to differences of opinion between reasonable researchers about which assumptions should be foundational. As a fellow researcher in vocal mimicry, I applaud the high quality of their scientific study and believe it represents an important contribution in the study of higher order sequences of animal vocalizations.

I am particularly intrigued by two aspects of the study. First, the authors demonstrate similarity of higher order sequences within and between individuals – an important result based on LSI and permutation test methods that could be applied to any sequence of animal vocalizations. The implication of the result for Albert's lyrebird is particularly interesting, i.e., that lyrebirds acquire their mimetic songs at least partially through social transmission from conspecifics rather than heterospecifics. This implication is not intuitive for researchers in vocal mimicry and represents an interesting puzzle that should stimulate future research. The second aspect of Backhouse et al's study that intrigues me is their result showing an acoustic contrast between subsequent song units, suggesting that recital mimicry functions to highlight to females the acoustic diversity of the signaler.

[We thank the reviewer for their time going through the revisions, and for the comments on the interesting aspects of this study. We also believe that the apparent social transmission of mimicry is interesting, particularly as vocal mimicry is often regarded as being purely acquired from heterospecifics.]

I have only one minor criticism, and it relates to the accuracy of the LSI measure in panel d of Fig. 1. (Thank you for fixing the LSI in panel e.) Comparing d to a, and after ignoring the whistle song, I see four changes (add A, ZàD, add K, and Rà3), with $1 - (4/17)$ not equal to 0.8. Please help me to understand your math.

[We thank the reviewer for pointing this out, and our sincere apologies for not fixing all mistakes the first time. It appears panel e was originally given the LSI for panel d by mistake, and the LSI for panel c was repeated for panel d. This has now been amended so all panels should have the correct corresponding LSIs.]

General comment #1 – my original concerns over immediate variety remain unresolved: I now see some value in how the authors rigorously quantified immediate variety, but the marginal benefit to their study still seems low. I'm also not convinced their methods would apply well beyond their specific study. Although Backhouse et al. adequately addressed my earlier criticisms, their justification in lines 175-177 seems to apply well only to sequences of vocal mimicry. If the sequence comes from a non-mimicking species (i.e., most species), or if the sequence comes from the non-mimetic song of a vocal mimic species, then it is impossible to assess whether something like vocalization F in Fig 1 would represent a single unit vs repetition of a single unit (element in this case). Even for mimicking species, the methods of Backhouse et al. might provide strange results for 'immediate' variety, e.g., if the mimic faithfully copies a model species that repeats an element for 20s or more. Given the authors already have a convincing result on acoustic contrast, I recommend dropping what seems like unnecessary data on immediate variety, but as I mentioned above, I do see some value in their approach and I am willing to yield to their opinion.

[We thank the reviewer for this perspective. We still believe that it is valuable to quantify immediate variety, as it is different to acoustic contrast - you can have immediate variety with low acoustic contrast, and you can have a high acoustic contrast with a higher than chance incidence of repeated units. In addition, most immediate variety in bird song is assumed rather than tested, and this method provides a way to quantify immediate variety, albeit with unit identification left to the researcher's discretion. Vocal mimicry certainly provides an easy way to delineate units, as units can be identified by their original use by the model species, and we have acknowledged this on lines 260-261 ("...we only tested for immediate repetition of vocal mimetic units, which may be objectively identified."). However,

units containing repeated elements may also be identified by, for example, silence surrounding the multi-element unit.

Given this reasoning we retain the analysis, however we have clarified our methods of unit identification to say that in this case, units containing repeated elements are short phrases, and therefore able to be treated as a single unit (lines 181 - 185: “Most mimetic units contained multiple elements, and some units were comprised of repetitions of short elements. These sequences of repeated elements were considered a single unit as they comprise short phrases and mimic how the vocalisation is sung by the model species.”)]

General comment #2 – definition of vocal mimicry: The authors rely heavily on the rigorous definition of vocal mimicry recommended in Dalziell et al. (2015), which focuses on function. The authors are correct that this definition is widely employed and consistent with theoretical explanations for other types of animal mimicry, such as Batesian mimicry. I nevertheless argue for a little caution about adopting a ‘function-only’ definition wholesale, given how little we know about vocal mimicry and given that vocal mimicry (as defined by Dalziell et al. 2015) occurs only in birds that learn their songs. Another way to define vocal mimicry focuses on development rather than function, and a developmental definition was apparently favored by at least Reviewer #2 and the editor (and myself for that matter). Another reason I urge caution is that the study of Backhouse et al. does not provide any direct evidence for function in the higher order song sequences they study. Instead, all their data focus on the behavior of senders. Without measuring the behavioral responses of receivers listening to the mimicry, the authors cannot clear the high bar set by the definition used by Dalziell et al. 2015. Overall, I don’t have a problem with Backhouse et al. using this definition in their paper, but it might be worth adding one or two caveats to the manuscript.

[This is a very interesting discussion point that we believe requires further attention, but is outside the scope of the current manuscript. It is true that we do not have direct evidence for the function of lyrebird mimicry. However, the acoustic fidelity of the mimicked sounds to the original model sounds suggests that there is selection acting on the accuracy of the mimicry. We also believe that socially transmitted mimicry should receive more attention in mimicry theory, particularly as Albert’s lyrebirds are unlikely to be the only species for which this is the case.

We have clarified our stance on using the term vocal mimicry in two places in the text. On lines 140 - 142 we now state: “In addition, while experimental testing would be required to determine the function of the recital mimicry, we make the assumption that recital mimicry is ‘functionally mimetic’ based on a high acoustic accuracy.” On lines 394 - 397 we state: “This social transmission of mimicry raises conceptual issues around what constitutes vocal mimicry. It is clear in this species that vocal mimicry can be acquired from both conspecifics and heterospecifics, and the role of conspecific tutors in mimicry acquisition further supports a definition of vocal mimicry that does not depend on the mode of acquisition.”]